# CAN TRANSFORMERS REASON LOGICALLY? A STUDY IN SAT SOLVING

## ABSTRACT

We theoretically and empirically study the logical reasoning capabilities of LLMs in the context of the Boolean satisfiability (SAT) problem. First, we construct a non-uniform class of decoder-only Transformers that can solve 3-SAT using backtracking and deduction via Chain-of-Thought (CoT). We prove its correctness by showing trace equivalence to the well-known DPLL SAT-solving algorithm. Second, to support the implementation of this abstract construction, we design a compiler `PARAT` that takes as input a procedural specification and outputs a transformer model implementing this specification. Third, rather than *programming* a transformer to reason, we evaluate empirically whether it can be *trained* to do so by learning directly from algorithmic traces ("reasoning paths") of the DPLL algorithm.

## 1 INTRODUCTION

Transformer-based Language Models (LLMs, Vaswani et al. (2017)) have demonstrated remarkable success in a wide range of tasks framed in natural language, especially when using prompting techniques such as Chain-of-Thought (CoT, Wei et al. (2022)). On the other hand, even the most advanced LLMs face challenges in reliable multi-step reasoning, frequently hallucinating towards nonsensical conclusions (Kambhampati et al. (2024)). Evaluating progress on logical deduction in language models remains an ongoing challenge as researchers have continued to disagree on even a reasonable definition of what constitutes "reasoning."

This paper focuses on the question of LLM reasoning capability in what we believe is the simplest and most mathematically precise setting: the Boolean satisfiability problem (SAT, Cook (1971)). SAT problems provide an excellent starting point for studying the reasoning ability of LLMs given that (a) natural language often encodes Boolean logic, and (b) we already have many useful algorithms that implement logical deduction to solve SAT problems Biere et al. (2009). Notably, notwithstanding the NP-completeness of SAT, humans implicitly solve simple boolean satisfaction problems in their daily lives; scheduling a multi-person meeting across time zones, for example.

In this work we aim to rigorously investigate Transformers' multi-step reasoning and backtracking capability in solving formal logical reasoning problems, and we demonstrate through a theoretical construction that decoder-only Transformers can reliably decide SAT instances.

**Theorem 1.1** (Informal version of Theorem 4.5). *For any $p, c \in \mathbb{N}^+$, there exist a decoder-only Transformer with $O(p^2)$ parameters that can decide all 3-SAT instances of at most $p$ variables and $c$ clauses using Chain-of-Thought reasoning.*

To investigate the properties of our construction empirically, we design a compiler that converts computational graphs of abstract sequence operations used in our construction into Transformer model weights. We implemented the construction in PyTorch and empirically validated its correctness on random 3-SAT instances. We also investigated its empirical properties such as the number of generated CoT tokens.

Additionally, we perform training experiments to demonstrate that Transformers can effectively learn from deductive reasoning and the backtracking process of the DPLL algorithm encoded as Chain-of-Thought. We show that Transformers equipped with CoT can generalize between SAT instances generated from different distributions within the same number of variables $p$. However,

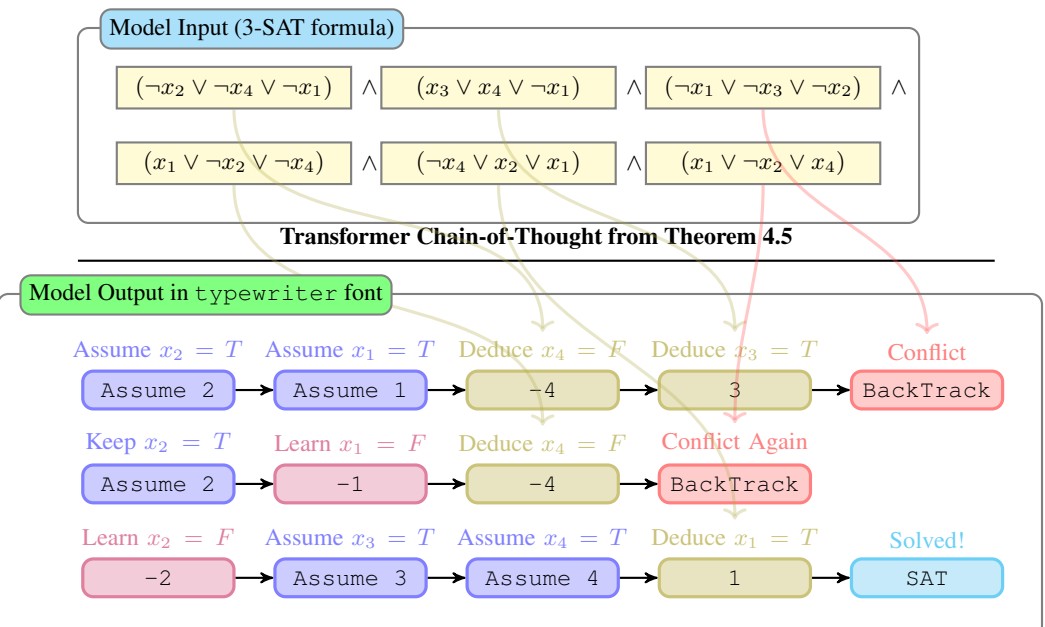

Figure 1: Visualization of the Chain-of-Thought (CoT) process used by our model to solve the SAT formula described in Theorem 4.5. The model autonomously performs trial-and-error reasoning, making multiple attempts and backtracking upon encountering conflicts. Here, $T$ represents *True* and $F$ represents *False*. Tokens in `typewriter` font denote the CoT generated by the model.

LLMs trained on SAT instances with CoT still struggle to solve instances with unseen number of variables, demonstrating challenges in learning length-generalizable reasoning and opportunities to incorporate compiled reasoning components in Transformer LLMs to improve reasoning capabilities.

**Contributions** We prove by theoretical construction that decoder-only Transformers can solve 3-SAT, a fundamental NP-Complete logical reasoning problem, by performing logical deduction and backtracking using Chain-of-Thought (CoT). We show that Transformers can perform logical deduction on all conditions (clauses) in parallel instead of checking each condition sequentially. Nevertheless, the construction requires exponentially many CoT steps in the worst case, although it is much faster on most typical examples.

We design `PARAT`, a compiler of high-level sequence operations written in Numpy-like syntax into Transformer model weights, to empirically validate and analyze theoretical constructions of Transformer algorithms.

We empirically demonstrate that the compiled SAT-solver model can solve SAT formulas up to 20 propositions and 88 clauses with perfect accuracy. Note that our goal is not to compete with modern state-of-the-art SAT solvers. Rather, we answer a fundamental question about whether LLMs can perform propositional reasoning with the 3-SAT problem. Finally, our training experiments suggest that Chain-of-Thought allows Transformer-LLMs to achieve out-of-distribution generalization for the same input lengths.

## 2 RELATED WORK

**Theoretical Expressiveness of Transformers and Chain-of-Thought (CoT):** Owing to the empirical success of Transformer-based models, many researchers have investigated the capabilities of the Transformer architecture from a theoretical perspective. This line of research focuses on what types of computation can Transformer models simulate by providing theoretical constructions of Transformer models with idealized assumptions. The seminal work of Liu et al. (2023) showed that Transformers can simulate automata using a single pass over only a logarithmic number of layers

w.r.t. the number of states. Yao et al. (2021) demonstrated that transformers can perform parentheses matching of at most $k$ types of parentheses and $D$ appearance of each ($\mathrm{Dyck}_{k,D}$) with $D + 1$ layers.

However, the computation power of one pass of the Transformer model is fundamentally limited (Merrill & Sabharwal (2023)), and the success of Chain-of-Thought (CoT) reasoning (Wei et al. (2022)) has sparked more recent research on how CoT can improve upon the expressiveness of Transformer models. Pérez et al. (2019) proved that Transformers can emulate the execution of single-tape Turing machines if each output vector is appended to the input vector sequence at the next iteration. Giannou et al. (2023) showed that Transformers can recurrently simulate arbitrary programs written in a one-instruction-set language if the output vector at every position of the Transformer is passed as input to the model at the next iteration. Li et al. (2024) proved that Transformers can simulate arbitrary boolean circuits using CoT by representing the circuit in the positional encoding and is commonly perceived to have shown that Transformers with CoT can "solve all problems". In particular, transformers can decide all problems in $\mathrm{P/poly} \supseteq \mathrm{P}$ with polynomial steps of CoT. Merrill & Sabharwal (2024) showed that Transformers with averaging hard attention can decide all regular languages with a linear number of CoT tokens and decide all problems in P with a polynomial number of CoT tokens. Feng et al. (2023) shows that Transformer CoT can perform integer arithmetic, solve linear equations, and perform dynamic programming for the longest increasing subsequence and edit distance problems. These seminal works profoundly advanced our understanding of the capabilities of Transformer models from a theoretical perspective.

**How our work differs from the above-mentioned results:** Many of the above papers are focused on problems in P or $\mathrm{P/poly}$, while 3-SAT is an NP-complete problem. It is widely believed that P is a strict subset of NP, and it is not known whether NP is a subset of $\mathrm{P/poly}$. In other words, our results are not comparable to these earlier results.

Meanwhile, Pérez et al. (2019), Li et al. (2024), and Merrill & Sabharwal (2024) also show that Transformers can simulate single-tape Turing Machines (TM) with CoT and can theoretically be extended to arbitrary decidable languages. However, these constructions require at least one CoT token for every step of TM execution. By contrast, our theoretical construction demonstrates that, for certain classes of formal reasoning problems, Transformers can simulate algorithmic reasoning traces at an abstract level with *drastically reduced number of CoT tokens* compared to step-wise emulation of a single-tape TM. At each CoT Step, our construction performs deductive reasoning over the full input in parallel while any single-tape TM must process each input token sequentially. Furthermore, the CoT produced by our theoretical construction abstractly represents the human reasoning process of trial and error, as demonstrated in Figure 1.

**Compilation of Transformer Weights.** Further, prior work on the theoretical construction of Transformer models rarely provide practical implementations. Notably, Giannou et al. (2023) provide an implementation of their construction and demonstrate its execution on several programs. However, the model is initialized "manually" using prolonged sequences of array assignments, limiting its extensibility to other theoretical frameworks.

More recently, Lindner et al. (2023) released Tracr, which compiles RASP (Weiss et al. (2021)) programs into decoder-only Transformer models. The "Restricted Access Sequence Processing Language" (RASP, Weiss et al. (2021)) is a human-readable representation of a subset of operations that Transformers can perform via self-attention and MLP layers. In our preliminary attempt to implement a SAT solver model with Tracr, we identified several implementation inconveniences and limitations of Tracr when scaling to more complex algorithms, which motivated the development of our compiler. In particular: (1) Every "variable" (termed `sop` in Lindner et al. (2023)) in Tracr must be either a one-hot categorical encoding or a single numerical value. This constraint makes representing more complex vector structures highly inconvenient. Furthermore, each `select` operation (i.e., self-attention) accepts only a single `sop` as the query and key vectors, whereas our theoretical construction often requires incorporating multiple `sops` as queries and keys. (2) Tracr represents position indices and many other discrete `sops` with a one-hot encoding, allocating a residual stream dimension for each possible value of the `sop`. In particular, compiling models with a context length of $n$ requires $O(n)$ additional embedding dimensions for each SOp that represents a position index. (3) For each binary operation between one-hot encoded `sops` (such as position indices), Tracr creates an MLP layer that first creates a lookup table of all possible value combinations of the input `sops`. This results in an MLP layer of $O(n^3)$ parameters.

## 3 PRELIMINARIES

The Boolean satisfiability problem (SAT) is the problem of determining whether there exists an assignment $A$ of the variables in a Boolean formula $F$ such that $F$ is true under $A$. In this paper we only consider 3-SAT instances in *conjunctive normal form* (CNF), where groups of at most 3 variables and their negations (*literals*) can be joined by OR operators into clauses, and these clauses can then be joined by AND operators. In our implementations we use the well-known *DIMACS* encoding for CNF formulae whereby each literal is converted to a positive or negative integer corresponding to its index, and clauses are separated by a 0 .

### 3.1 AUTOREGRESSIVE DECODER-ONLY TRANSFORMER ARCHITECTURE

The Transformer architecture Vaswani et al. (2017) is a foundational model in deep learning for sequence modeling tasks. In our work, we focus on the autoregressive decoder-only Transformer, which generates sequences by predicting the next token based on previously generated tokens. It is a relatively complex architecture, and here we only give a precise but quite concise description, and we refer the reader Vaswani et al. (2017) among many others for additional details. Given an input sequence of tokens $\mathbf{s} = (s_1, s_2, \ldots, s_n) \in \mathcal{V}^n$, where $\mathcal{V}$ is a *vocabulary*, a Transformer model $M : \mathcal{V}^* \to \mathcal{V}$ maps $\mathbf{s}$ to an output token $s_{n+1} \in \mathcal{V}$ by composing a sequence of parameterized intermediate operations. These begin with a token embedding layer, following by $L$ *transformer blocks* (*layers*), each block consisting of $H$ *attention heads*, with embedding dimension $d_{\text{emb}}$, head dimension $d_h$, and MLP hidden dimension $d_{\text{mlp}}$. Let us now describe each of these maps in detail.

**Token Embedding and Positional Encoding.** Each input token $s_i$ is converted into a continuous vector representation $\text{Embed}(s_i) \in \mathbb{R}^d$ using a fixed embedding map $\text{Embed}(\cdot)$. To incorporate positional information, a positional encoding vector $\boldsymbol{p}_i \in \mathbb{R}^d$ is added to each token embedding. The initial input to the first Transformer block is

$$\boldsymbol{x}^{(0)} \leftarrow (\text{Embed}(s_1) + \boldsymbol{p}_1, \ \text{Embed}(s_2) + \boldsymbol{p}_2, \ \ldots, \ \text{Embed}(s_n) + \boldsymbol{p}_n) \in \mathbb{R}^{n \times d}.$$

**Transformer Blocks.** For $l = 1, \ldots, L$, each block $l$ of the transformer processes an embedded sequence $\boldsymbol{x}^{(l-1)} \in \mathbb{R}^{n \times d}$ to produce another embedded sequence $\boldsymbol{x}^{(l)} \in \mathbb{R}^{n \times d}$. Each block consists of a multi-head self-attention (MHA) mechanism and a position-wise feed-forward network (MLP). We have a set of parameter tensors that includes MLP parameters $\boldsymbol{W}_1^{(l)} \in \mathbb{R}^{d_{\text{emb}} \times d_{\text{mlp}}^*}$, $\boldsymbol{b}_1^{(l)} \in \mathbb{R}^{d_{\text{mlp}}^*}$, $\boldsymbol{W}_2^{(l)} \in \mathbb{R}^{d_{\text{mlp}} \times d}$, and $\boldsymbol{b}_2^{(l)} \in \mathbb{R}^d$, self-attention parameters $\boldsymbol{W}_Q^{(l,h)}$, $\boldsymbol{W}_K^{(l,h)}$, $\boldsymbol{W}_V^{(l,h)} \in \mathbb{R}^{d \times d_h}$ for every $h = 1, \ldots, H$, and multi-head projection matrix $\boldsymbol{W}_O^{(l)} \in \mathbb{R}^{(H d_h) \times d_{\text{emb}}}$. We will collectively refer to all such parameters at layer $l$ as $\Gamma^{(l)}$, whereas the self-attention parameters for attention head $h$ at layer $l$ will be referred to as $\Gamma^{(l,h)}$. We can now process the embedded sequence $\boldsymbol{x}^{(l-1)}$ to obtain $\boldsymbol{x}^{(l)}$ in two stages:

$$\boldsymbol{h}^{(l)} \leftarrow \boldsymbol{x}^{(l-1)} + \text{MHA}\left(\boldsymbol{x}^{(l-1)}; \Gamma^{(l)}\right), \qquad \text{and} \qquad \boldsymbol{x}^{(l)} \leftarrow \boldsymbol{h}^{(l)} + \text{MLP}\left(\boldsymbol{h}^{(l)}; \Gamma^{(l)}\right),$$

where

$$\text{MHA}\left(\boldsymbol{x}; \Gamma^{(l)}\right) := \text{Concat}\left(\text{Attention}(\boldsymbol{x}; \Gamma^{(l,1)}), \ldots, \text{Attention}(\boldsymbol{x}; \Gamma^{(l,H)})\right) \boldsymbol{W}_O^{(l)}$$

$$\text{Attention}(\boldsymbol{x}; \Gamma^{(l,h)}) := \text{softmax}\left(d_h^{-1/2} \boldsymbol{x} \boldsymbol{W}_Q^{(l,h)} (\boldsymbol{W}_K^{(l,h)} \boldsymbol{x})^\top + \boldsymbol{M}\right) \boldsymbol{x} \boldsymbol{W}_V^{(l,h)}$$

$$\text{MLP}\left(\boldsymbol{h}; \Gamma^{(l)}\right) := \sigma\left(\boldsymbol{h} \boldsymbol{W}_1^{(l)} + \boldsymbol{b}_1^{(l)}\right) \boldsymbol{W}_2^{(l)} + \boldsymbol{b}_2^{(l)}.$$

The $n \times n$ matrix $\boldsymbol{M}$ is used as a "mask" to ensure self-attention is only backward looking, so we set $\boldsymbol{M}[i, j] = \infty$ for $i \geq j$ and $\boldsymbol{M}[i, j] = 0$ otherwise. Finally, we use the $\text{ReGLU}(\cdot) : \mathbb{R}^{2d_{\text{mlp}}} \to \mathbb{R}^{d_{\text{mlp}}}$ activation function $\sigma(\cdot)$ at each position. Tiven input $\boldsymbol{u} \in \mathbb{R}^{n \times 2d_{\text{mlp}}}$, for each position $i$ we split $\boldsymbol{u}_i$ into two halves $\boldsymbol{u}_{i,1}, \ \boldsymbol{u}_{i,2} \in \mathbb{R}^d$ and, using $\otimes$ denotes element-wise multiplication, we define

$$\sigma_{\text{ReGLU}}\left(\boldsymbol{u}_i\right) = \boldsymbol{u}_{i,1} \otimes \text{ReLU}\left(\boldsymbol{u}_{i,2}\right). \tag{1}$$

**Output Layer.** After the final Transformer block, the output representations are projected onto the vocabulary space to obtain a score for each token. We assume that we're using the greedy decoding strategy, where the token with the highest score at the last input position is the model output.

---

**Algorithm 1:** Greedy Decoding

---

**Input:** Model $M : \mathcal{V}^* \to \mathcal{V}$, prompt $\boldsymbol{s}_{1:n} = (s_1, s_2, \ldots, s_n)$, stop tokens $\mathcal{E} \subseteq \mathcal{V}$, $t \leftarrow n$

1 **while** $t \leftarrow t + 1$ **do**
2     $s_t \leftarrow M(\boldsymbol{s}_{1:t-1})$ ;         // Obtain model output and append to string
3     **if** $s_t \in \mathcal{E}$ **return** $\boldsymbol{s}_{1:t}$
4 **end**

---

$$\boldsymbol{o} = \boldsymbol{x}^{(L)} \boldsymbol{W}_{\text{out}} + \boldsymbol{b}_{\text{out}} \in \mathbb{R}^{n \times V}, s_{n+1} = \arg\max_v \boldsymbol{o}_{n,v} \in \mathcal{V} \tag{2}$$

where $\boldsymbol{W}_{\text{out}} \in \mathbb{R}^{d \times V}$, $\boldsymbol{b}_{\text{out}} \in \mathbb{R}^V$, $V$ is the size of the vocabulary, $\boldsymbol{o}_{n,v}$ is the score for token $v$ at the last input position $n$.

**Autoregressive Decoding and Chain-of-Thought.** During generation, the Transformer model is repeatedly invoked to generate the next token and appended to the input tokens, described in Algorithm 1. In this paper, we refer to the full generated sequence of tokens as the **Chain-of-Thought**, and the number of chain-of-thought tokens in Algorithm 1 is $t - n$.

## 4 TRANSFORMERS AND SAT: LOGICAL DEDUCTION AND BACKTRACKING

This section presents and explains our main results on Transformers' capability in deductive reasoning and backtracking with CoT. To rigorously state our results, we first formally define decision problems, decision procedures, and what it means for a model to "solve" a decision problem using CoT:

**Definition 4.1** (Decision Problem). Let $\mathcal{V}$ be a vocabulary, $\Sigma \subseteq \mathcal{V}$ be an alphabet, $L \subseteq \Sigma^*$ be a set of valid input strings. We say that a mapping $f : L \to \{0, 1\}$ is a *decision problem* defined on $L$.

**Definition 4.2** (Decision Procedure). We say that an algorithm $\mathcal{A}$ is a decision procedure for the decision problem $f$, if given any input string $x$ from $L$, $\mathcal{A}$ halts and outputs 1 if $f(x) = 1$, and halts and outputs 0 if $f(x) = 0$.

**Definition 4.3** (Autoregressive Decision Procedure). For any map $M : \mathcal{V}^* \to \mathcal{V}$, which we refer to as an *auto-regressive next-token prediction model*, and $\mathcal{E} = \{\mathcal{E}_0, \mathcal{E}_1\} \subset \mathcal{V}$, define procedure $\mathcal{A}_{M,\mathcal{E}}$ as follows: For any input $s_{1:n}$, run Algorithm 1 with stop tokens $\mathcal{E}$. $\mathcal{A}_{M,\mathcal{E}}$ outputs 0 if $s_{1:t}$ ends with $\mathcal{E}_0$ and $\mathcal{A}_{M,\mathcal{E}}$ output 1 otherwise. We say $M$ *autoregressively decides* decision problem $f$ if there is some $\mathcal{E} \subset \mathcal{V}$ for which $\mathcal{A}_{M,\mathcal{E}}$ decides $f$.

**Definition 4.4** (3-SAT$_{p,c}$). Let DIMACS$(p, c)$ denote the set of valid DIMACS encodings of 3-SAT instances with at most $p$ variables and $c$ clauses with a prepended [BOS] token and an appended [SEP] token. Define 3-SAT$_{p,c}$ : DIMACS$(p, c) \to \{0, 1\}$ as the problem of deciding whether the 3-SAT formula encoded in the input in DIMACS$(p, c)$ encoding is satisfiable.

With the above definition, we're ready to present a formal statement of our theoretical construction of a Transformer model that performs SAT Solving:

**Theorem 4.5** (Decoder-only Transformers can solve SAT). *For any $p, c \in \mathbb{N}^+$, there exists a Transformer model $M : \mathcal{V}^* \to \mathcal{V}$ that autoregressively decides 3-SAT$_{p,c}$ in no more than $p \cdot 2^{p+1}$ CoT iterations. $M$ requires $L = 7$ layers, $H = 5$ heads, $d_{emb} = O(p)$, and $O(p^2)$ parameters.*

Remarks on Theorem 4.5

- The upper bound on the CoT length $p \cdot 2^{p+1}$ is a worst-case upper bound which assumes that the model is unable to make any logical deductions have to try all $2^p$ assignments. However, this upper bound is never reached in practice, and in Figure 4 we show that the number of CoT tokens is no greater than $8p \cdot 2^{0.08p}$ for most formulas. If the number of backtracking steps is bounded by $T$ then the CoT is no longer than $(2p + 1)(T + 1)$

- The worst-case CoT length is independent of the number of clauses $c$, which is due to the parallel deduction over all clauses within the Transformer construction. Otherwise, sequentially processing each clause would take at least $c \cdot 2^{O(p)}$ number of steps.

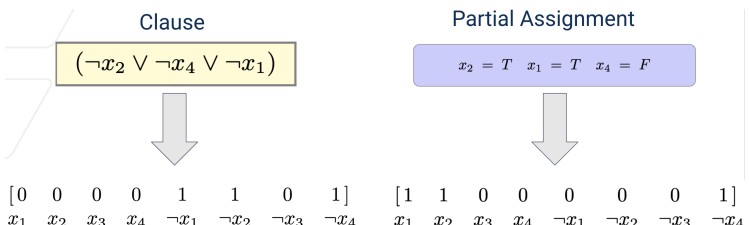

Figure 2: Illustration of the encoding scheme $E(C)$ and $E(A)$ for clauses and partial assignments from Definition 4.6 and Definition 4.7 with $p = 4$ varables.

- Positional encodings are not included in the number of parameters. The positional encoding at position $i$ is the numerical value $i$ at a particular dimension.
- Each parameter can be represented with $O(p + \log c)$ bits

We show our full proof via trace equivalence with abstract DPLL (Nieuwenhuis et al. (2005)) in Appendix C. The construction uses adapted versions of lemmas from Feng et al. (2023) as basic building blocks. Here we provide a proof sketch of the core operations in our theoretical construction.

**Proof Sketch** In Figure 1 we illustrate the CoT process used by our theoretical construction, which uses CoT tokens to simulate various operations including unit propagation (i.e., logical deduction), variable decision, and backtracking.

To process clauses and partial assignments with attention operations, the initial layers of the theoretical construction compute the binary vector encodings of clauses and partial assignments and store them in the hidden states. We formally define the encoding scheme for clauses and partial assignments below:

**Definition 4.6** (Encoding of clause). Let $C$ be a clause. Define encoding $E(C) \in \{0, 1\}^{2p}$ of clause $C$ as the following: For $v \in [p]$, $E(C)_v = 1$ iff $x_v$ is a literal in $C$, and $E(C)_{p+v} = 1$ iff $\neg x_v$ is a literal in $C$. All positions in $E(C)$ are 0 otherwise.

**Definition 4.7** (Encoding of partial assignment). Let $A : \{x_1, \ldots, x_p\} \to \{\text{True}, \text{False}, \text{None}\}$ be a partial assignment. Define encoding $E(A) \in \{0, 1\}^{2p}$ of clause $C$ as the following: For $v \in [p]$, $E(A)_v = 1$ iff $A(x_v) = \text{True}$, and $E(A)_{p+v} = 1$ iff $A(x_v) = \text{False}$. All positions in $E(C)$ are 0 otherwise.

We also define a variant of the partial assignment encoding as an affine Transformers of $E(A)$, which sets both positions corresponding to a variable to 1 if the variable is unassigned:

**Proposition 4.8.** *For partial assignment $A$, define $E_{\text{not-false}}(A) = M_{\text{not-false}} \cdot E(A) + \mathbf{1}^{2p}$ where*

$$M_{\text{not-false}} = \begin{bmatrix} \mathbf{0} & -\mathbf{I}_p \\ -\mathbf{I}_p & \mathbf{0} \end{bmatrix} \in \mathbb{R}^{2p \times 2p} \text{ and } \mathbf{1}^{2p} \text{ is the all ones vector. Then, for } v \in [p]:$$

$$E_{\text{not-false}}(A)_v = 1 \text{ iff } A(x_v) \in \{\text{True}, \text{None}\}, \quad E_{\text{not-false}}(A)_{p+v} = 1 \text{ iff } A(x_v) \in \{\text{False}, \text{None}\}.$$

We now show that the relationship between a 3-SAT formula and a partial assignment can be established using their binary encoding:

**Lemma 4.9.** *Let $F$ be a 3-SAT formula over variables $\{x_1, \ldots, x_p\}$ with $c$ clauses $\{C_1, \ldots, C_c\}$ and $A$ a partial assignment defined on variables $\{x_1, \ldots, x_p\}$, then the following properties hold:*

1. *Satisfiability Checking: The partial assignment $A$ satisfies the formula $F$ if and only if:*

$$\forall i \in [c], \quad E(C_i) \cdot E(A) \geq 1.$$

2. *Conflict Detection: The partial assignment $A$ contradicts the formula $F$ if and only if:*

$$\exists i \in [c], \quad E(C_i) \cdot E_{\text{not-false}}(A) = 0.$$

3. *Deduction: If partial assignment $A$ does not contradict formula $F$, then*

*(a) A variable $x_v$ is implied to be* true *under A and F if:*

$$\exists i \in [c], \quad E(C_i) \cdot E_{\text{not-false}}(A) \leq 1, \quad E(C_i)_v = 1, \quad and \quad E(A)_v = E(A)_{p+v} = 0.$$

*(b) A variable $x_v$ is implied to be* false *under A and F if :*

$$\exists i \in [c], \quad E(C_i) \cdot E_{\text{not-false}}(A) \leq 1, \quad E(C_i)_{p+v} = 1, \quad and \quad E(A)_v = E(A)_{p+v} = 0.$$

Recall that an attention head computes a query and key vector from the hidden states and the attention weight between two positions is based on the dot product between the query vector of the source position and the key vector of the target position. If the Transformer weights are configured such that the query vectors are Figure 1 is $E(A)$ or $E_{\text{not-false}}(A)$ for partial assignments $A$ in the Chain-of-Thought illustrated, and the key vectors are $E(C_i)$ for positions of clauses $C_i$ in the formula, then the attention weight (before softmax) would be proportional to $E(C_i) \cdot E(A)$ or $E(C_i) \cdot E_{\text{not-false}}(A)$ respectively, which are values crucial for the operations in 4.9. We can then scale the attention weights so that the the attention weights focus on only the extremal values of $E(C_i) \cdot E(A)$ or $E(C_i) \cdot E_{\text{not-false}}(A)$. We illustrate the consequence of this correlation with the following informal lemma, which considers an idealized input that contains only the positions with encoding vectors and auxiliary values:

**Lemma 4.10** (Parallel Processing of Clauses, Informal). *Let F be a 3-SAT formula over variables $\{x_1, \ldots, x_p\}$ with c clauses $\{C_1, \ldots, C_c\}$ and A a partial assignment defined on variables $\{x_1, \ldots, x_p\}$. Let*

$$X_{encoding} = \begin{bmatrix} 0 & 1 & 1 \\ E(C_1) & 0 & 1 \\ \vdots & \vdots & \vdots \\ E(C_c) & 0 & 1 \\ E(A) & 0 & 1 \end{bmatrix} \in \mathbb{R}^{(c+2) \times (2p+2)}$$

*which includes encoding of clauses in F and partial assignment A as well as added auxiliary values. Let $\mathbf{1}_{A \models F}$ denote the indicator variable of whether A satisfy formula F, $\mathbf{1}_{A \not\models F}$ denote the indicator variable of whether A constradict F, and $e_{UP} \in 0, 1^{2p}$ denote the encoding of all variable assignments that can be deduced from A and F, then with $X_{encoding}$ as input and any $1 > \epsilon > 0$ there exists:*

- *An attention head that outputs $\mathbf{1}_{A \models F}$ with approximation error bounded by $\epsilon$*

- *An attention head that outputs $\mathbf{1}_{A \not\models F}$ with approximation error bounded by $\epsilon$*

- *An attention head followed by a MLP layer that outputs $e_{UP}$ with $\|\cdot\|_\infty$ error bounded by $\epsilon$*

*and all weight values are bounded by $O(poly(p, c, \log(1/\epsilon)))$*

Lemma 4.10 essentially shows that, when given the binary encoding of clauses and a partial assignment, a single Transformer layer can perform satisfiability checking, conflict detection, and deduction over all clauses in the formula in parallel, which is the core reasoning our theoretical construction uses drastically less CoT tokens than step-wise simulation of Turing Machines.

The remaining parts of the construction performs indexing operations that translates DIMACS encodings into our encoding of clauses and partial assignments and selects the correct output token from the results of the operations described in Lemma 4.10.

## 5 Compiler for Complex Transformer Algorithms

In the previous section, we presented a theoretical construction of a Transformer capable of solving SAT instances through backtracking and parallel deduction. However, relying solely on theorems and proofs can make it challenging to gain practical insights and verify correctness. To address this, we introduce ParametricTransformer compiler and the corresponding PARAT language, which provides a framework for converting theoretical constructions of Transformers into practical models to facilitate empirical analysis and validation.

The syntax of the `PARAT` **language** is a restricted subset of Python with the NumPy library. Every variable `v` in `PARAT` is a 2-D NumPy array of shape `n x d_v`, where `n` denotes the input number of tokens and `d_v` is the dimension of the `PARAT` variable `v`, which can be different for every variable `v`.

A program in the `PARAT` language is composed of a linear sequence of statements (i.e., no control flow such as loops or branching is allowed), where each statement assigns the value of an expression to a variable. Let `v_1`, `v_2`, ... denote `PARAT` variable names. Then, each statement involving `PARAT` variables must be one of the following:

- **Binary operations:** `v_1 + v_2`, `v_1 * v_2`, `v_1 - v_2`
- **Index operations:** `v_1[v_2, :]`, `v_1[:, start:end]`, where `start` and `end` are non-negative integers
- **Function calls:** A function from our predefined library of functions that takes as input `PARAT` variables

The input variables of a PARAT program for vocabulary size $V$ are `tokens` and `indices`, where `tokens` is a $V$-dimensional `PARAT` variable containing one-hot token embeddings of the input tokens, and `indices` is a 1-dimensional `PARAT` variable containing the numerical index of each input token (i.e., the array `[[1], [2], ..., [n]]`).

The ParametricTransformer **compiler** takes in a program written in the PARAT language and a PARAT variable `out` of dimension $V$ and outputs a PyTorch `Module` object that implements a Transformer model as defined in Section 2. The following condition is satisfied: For any possible input sequence of tokens $s$ in the vocabulary of length $n$, the token predicted by the Transformer model is the same as the token corresponding to `out[-1, :].argmax()` (i.e., the token prediction at the last position) when interpreting the PARAT program using the Python interpreter with the NumPy library.

## 5.1 ANALYSIS OF THE COMPILED SAT-SOLVING MODEL

With our compiler, we successfully compiled our theoretical construction in Theorem 4.5 using the code in Appendix D. For $p = 20$ number of variables, the resulting Transformer has 7 layers, 5 attention heads, 502 embedding dimensions, and 5011862 parameters. With a concrete implementation of our theoretical construction in PyTorch, we empirically investigate 3 questions (1) Does the compiled model correctly decide SAT instances? (2) How many steps does the model take to solve actual 3-SAT instances? (3) How does error induced by soft attention affect reasoning accuracy? These questions reveal further insights that are not available by observing the theoretical constructions alone and demonstrate the additional values provided by `PARAT`.

**Evaluation Datasets** We evaluate our models on randomly sampled DIMACS encoding of 3-SAT formulas. We focus on SAT formulas with exactly 3 literals in each clause, with the number of clauses $c$ between $4.1p$ and $4.4p$, where $p$ is the number of variables.

It is well-known that the satisfiability of such random 3-SAT formulas highly depends on the clause/variable ratio, where a formula is very likely satisfiable if $c/p \ll 4.26$ and unsatisfiable if $c/p \gg 4.26$ (Crawford & Auton (1996)). This potentially allows a model to obtain high accuracy just by observing the statistical properties such as the $c/p$ ratio. To address this, we constrain this ratio for all formulas to be near the critical ratio $4.26$. Furthermore, our "marginal" datasets contain pairs of SAT vs UNSAT formulas that differ from each other by only a single literal. This means that the SAT and UNSAT formulas in the dataset have almost no statistical difference in terms of $c/p$ ratio, variable distribution, etc., ruling out the possibility of obtaining SAT vs UNSAT information solely via statistical properties.

We also use 3 different sampling methods to generate formulas of different solving difficulties to evaluate our model:

- **Marginal:** Composed of pairs of formulas that differ by only one token.
- **Random:** Formulas are not paired by differing tokens and each clause is randomly generated.
- **Skewed:** Formulas where polarity and variable sampling are not uniform; For each literal, one polarity is preferred over the other. Some literals are also preferred over others.

We generate the above 3 datasets for each variable number $4 \leq p \leq 20$, resulting in 51 total datasets of 2000 samples each. Each sample with $p$ variables contains $16.4p$ to $17.6p$ input tokens, which is at least 320 for $p = 20$.

**Model** Unless otherwise stated, the model we experiment with is compiled from the code in D using `PARAT` with max number of variables $p = 20$, max number of clauses $c = 88$, and exactness parameter $\beta = 20$. The model uses greedy decoding during generation.

**Accuracy** Our compiled model achieves perfect accuracy on all evaluation datasets described above. This provides empirical justification for our theoretical construction for Theorem 4.5 as well as `PARAT`. This result is included in Figure 3 to compare with trained models.

**How many steps?** We perform experiments to measure the empirical Chain-of-Thought length required for solving SAT formulas of different sizes. For all formulas we evaluated, the maximum CoT length is bounded by $8p \cdot 2^{0.08p}$, which is significantly less than the theoretical bound of $p \cdot 2^{(p+1)}$. This indicates that the model can use deduction to reduce the search space significantly. The figure illustrating the results is in Appendix Figure 4.

**Effect of Soft Attention** In our previous evaluations, we used a sufficiently large "exactness" value $\beta$ to ensure that the error from `MEAN` based operations does not affect the final output of greedy sampling. The use of "Averaging Hard Attention" is prevalent in previous works on theoretical construction. However, how exactly does soft-attention affect the final reasoning output?

In Figure 5 we present the SAT/UNSAT prediction accuracy for models under 8 different "mean exactness" $\beta$ values on our "marginal" datasets ranging from 2.5 to 20. Recall that $\beta$ controls how the well soft attention approximates "hard" attention in each self-attention layer. Our results demonstrate that longer inputs generally require larger $\beta$ values to achieve high accuracy. This may explain why Transformers fail to learn generalizable algorithmic procedures, as the attention learned on smaller formulas may be too "soft" to generalize to larger inputs.

# 6 CAN TRANSFORMER LEARN SAT SOLVING FROM DATA?

Our previous sections showed that Transformer and weights exist for solving SAT instances using CoT with backtracking and deduction. However, it is unclear to what extent Transformers can learn such formal reasoning procedures by training on SAT formulas. Previously, Zhang et al. (2023) showed that when using a single pass of a Transformer model (without CoT), Transformers fail to generalize to logical puzzles sampled from different distributions even when they have the same number of propositions.

This section provides proof-of-concept evidence that training on the Chain-of-Thought procedure with deduction and backtracking described in Figure 1 can facilitate Out-of-Distribution generalization within the same number of variables.

**Datasets** In Section 5.1 we introduced 3 different distributions over random 3-SAT formulas of varying difficulties. For training data, we use the same sampling methods, but instead of having a separate dataset for each variable number $p$, we pick 2 ranges $p \in [6, 10]$ and $p \in [11, 15]$, where for each sample a random $p$ value is picked uniformly random from the range. Each formula with $p$ variables contains $16.4p$ to $17.6p$ tokens. This results in $2 \times 3$ training datasets, each containing $5 \times 10^5$ training samples[1], with balanced SAT vs UNSAT samples. For each formula, we generate the corresponding chain of thought in the same format as Figure 1 using a custom SAT Solver. The evaluation data is exactly the same as Section 5.1.

**Model and Training** We use the LLaMa (Touvron et al. (2023)) architecture with 70M and 160M parameters for the training experiments, which uses Rotary Positional Encodings (RoPE) and SwiGLU as the activation function for MLP layers. Following prior works (Feng et al. (2023)), we compute cross-entropy loss on every token in the CoT but not the DIMACS encoding in the prompt tokens. We provide further training details in Appendix A. We also permute the variable IDs for training samples to ensure that the model sees all possible input tokens for up to 20 variables.

---

[1]The number of training samples is negligible compared to the total number of possible formulas. Note that the number of clauses is at least $4p$, each clause contains 3 literals and each literal has at least $p$ choices. This results in $p^{12p}$ possibilities, which is $> 10^{56}$ for $p = 6$

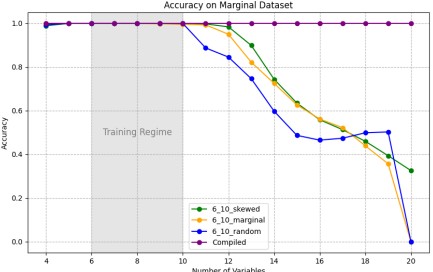 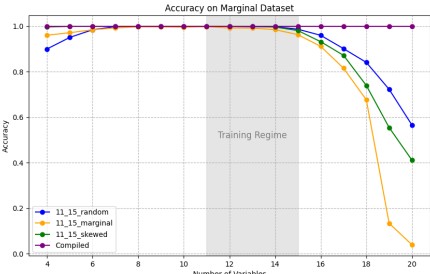

Figure 3: Result of the Length generalization experiments, showing SAT/UNSAT prediction accuracy of Transformer-LLM trained on the marginal, random, and skewed dataset on the marginal dataset over 4-20 variables. Left: model trained on 6-10 variables. Right: model trained on 11-15 variables.

## 6.1 INTRA-LENGTH OOD GENERALIZATION

Table 1: Average accuracies (%) of SAT/UNSAT prediction for models trained and tested on different datasets in the training regime for number of variables $p \in [6, 10]$ and $p \in [11, 15]$. Columns denote train datasets, and rows denote test datasets. Each accuracy is computed over 10000 total samples.

|  | $p \in [6, 10]$ | | | $p \in [11, 15]$ | | |
|---|---|---|---|---|---|---|
|  | Marginal | Random | Skewed | Marginal | Random | Skewed |
| **Marginal** | 99.88% | 99.99% | 99.99% | 98.66% | 99.70% | 99.57% |
| **Random** | 99.96% | 100.00% | 100.00% | 99.11% | 99.75% | 99.55% |
| **Skewed** | 99.96% | 100.00% | 99.99% | 99.41% | 99.74% | 99.48% |

Our first set of experiments evaluates the model's performance on SAT formulas sampled from different distributions from training, but the number of variables in formulas remains the same ($p \in [6, 10]$ and $p \in [11, 15]$ for both train and test datasets).

As shown in Table 1, our trained models achieve near-perfect SAT vs UNSAT prediction accuracy when tested on the same number of variables as the training data, even when on formulas sampled from different distributions. Recall that the "marginal" dataset has SAT vs UNSAT samples differing by a single token (out of at least $16p$ tokens in the input formula), which minimizes statistical evidence that can be used for SAT/UNSAT prediction. Our experiments suggest that the LLM have very likely learned general reasoning procedures using CoT that can be applied to all formulas with the same number of variables as the data they are trained on.

## 6.2 LIMITATIONS IN LENGTH GENERALIZATION

The second experiment evaluates the model's ability to generalize to formulas with a different number of variables than seen during training. We use the model trained on 3 data distributions described in section 6.1 and evaluate the marginal dataset with 4-20 variables, generated using the three methods described, with 2,000 samples each. For this experiment, we evaluate the accuracy of the binary SAT vs UNSAT prediction.

**Results** In Figure 3, our results indicate that performance degrades drastically beyond the training regime when the number of variables increases. This shows that the model is unable to learn a general SAT-solving algorithm that works for all inputs of arbitrary lengths, which corroborates our theoretical result where the size of the Transformer for SAT-solving depends on the number of variables. This further demonstrates the value of having a compiled Transformer that provably works well on all inputs up to $p$ variables for any given $p$.

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

## A TRAINING DETAILS

We use Llama Touvron et al. (2023) models in the HuggingFace library. For the 70M model, we use models with 6 layers, 512 embedding dimensions, 8 heads, 512 attention hidden dimensions, and 2048 MLP hidden dimensions. For the 140M model, we use 12 layers, 768 embedding dimensions, 12 heads, 768 attention hidden dimensions, and 3072 MLP hidden dimensions. Both models have 850 context size. We trained for 5 epochs on both datasets using the Adam optimizer with a scheduled cosine learning rate decaying from $6 \times 10^{-4}$ to $6 \times 10^{-5}$ with $\beta_1 = 0.9$ and $\beta_2 = 0.95$.

## B ADDITIONAL EXPERIMENT RESULTS

In Figure 4 we provide results on the number of Chain-of-Thought tokens required to solve randomly generated SAT instances. In Figure 5 we provide results on how the SAT/UNSAT prediction accuracy is affected by numerical errors introduced by softmax.

## C PROOFS

### C.1 NOTATION DETAILS

**3-SAT** SAT problems where the Boolean formula is expressed in conjunctive normal form (CNF) with three literals per clause will be referred to as *3-SAT*. A formula in CNF is a conjunction (i.e. "AND") of clauses, a **clause** is a disjunction (i.e. "OR") of several **literals**, and each literal is either a variable or its negation. In the case of 3-SAT, each clause contains at most three literals. An example 3-SAT formula with 4 variables and 6 clauses is:

$$(x_1 \vee \neg x_2) \wedge (\neg x_1 \vee x_2 \vee \neg x_3) \wedge (x_2 \vee x_4 \vee \neg x_1) \wedge$$
$$(x_1 \vee \neg x_3 \vee x_4) \wedge (\neg x_2 \vee \neg x_3 \vee \neg x_4) \wedge (\neg x_4 \vee \neg x_1)$$

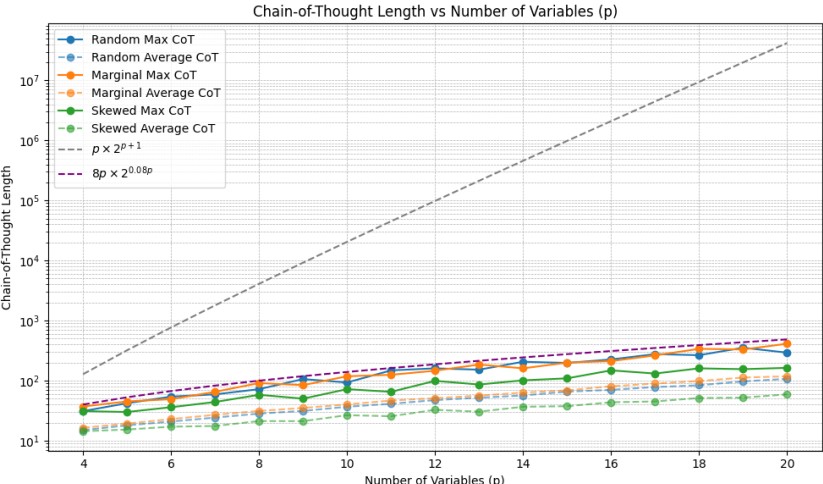

Figure 4: Chain-of-Thought Lengths generated by the compiled SAT-Solver Model vs the number of boolean variables in sampled SAT formulas, y-axis in log scale. Solid lines denote the maximum CoT length for each dataset while opaque, dashed lines denote the average CoT length. The empirical maximum CoT length in our datasets is bounded by $8p \cdot 2^{0.08p}$
.

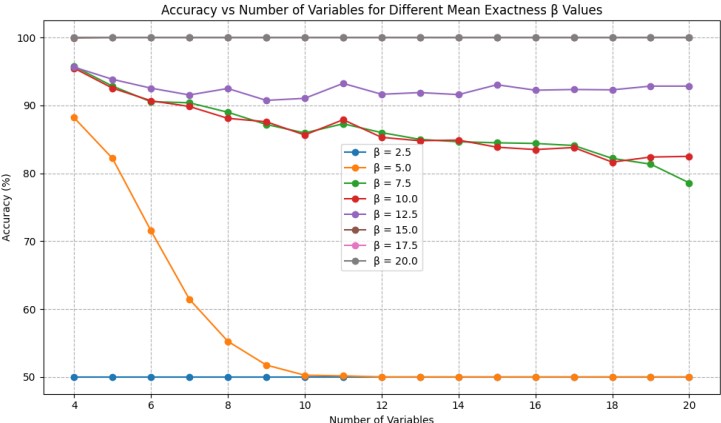

Figure 5: The impact of soft attention in Transformer layers on the SAT/UNSAT prediction accuracy. $\beta$ is a scaling factor that allows the soft attention operation to better simulate hard attention at the cost of larger model parameter values in attention layers. The model achieves perfect accuracy on all "marginal" datasets starting at $\beta = 17.5$, and for lower $\beta$ values, accuracy is negatively correlated with the number of variables in the datasets.

In the above formula, $(x_1 \lor \neg x_2)$ is a clause, which contains the literals $x_1$ and $\neg x_2$.

The 3-SAT problem refers to determining if any assignment of truth values to the variables allows the formula $\phi$ to evaluate as true. It is well-known that 3-SAT is NP-hard and is widely believed to be unsolvable in polynomial time.

**DIMACS Encoding** The DIMACS format is a standardized encoding scheme for representing Boolean formulas in conjunctive normal form (CNF) for SAT problems. Each clause in the formula is represented as a sequence of integers followed by a terminating "0" (i.e. "0" represents $\land$ symbols and parentheses). Positive integers correspond to variables, while negative integers represent the negations of variables. For instance, if a clause includes the literals $x_1$, $\neg x_2$, and $x_3$, it would be represented as "`1 -2 3 0`" in the DIMACS format.

For the 3-SAT example in the previous paragraph, the corresponding DIMACS representation would be:

```
1 -2 0 -1 2 -3 0 2 4 -1 0 1 -3 4 0 -2 -3 -4 0 -4 -1 0
```

## C.2 USEFUL LEMMAS FOR TRANSFORMERS

In this section, we present adapted versions of several lemmas from Feng et al. (2023). Specifically, an MLP with ReGLU can exactly simulate ReLU, linear operations, and multiplication without error. For Self-attention lemmas, we directly adapt from Feng et al. (2023).

### C.2.1 LEMMAS FOR MLP WITH REGLU ACTIVATION

This section shows several lemmas showing the capabilities of the self-attention operation and MLP layers to approximate high-level vector operations. These high-level operations are later used as building blocks for the Transformer SAT-solver. Specifically, with appropriate weight configurations, a 2-layer MLP with ReGLU activation $f(\boldsymbol{x}) = \boldsymbol{W}_2[(\boldsymbol{W}_1\boldsymbol{x} + \boldsymbol{b}) \otimes \mathrm{relu}(\boldsymbol{V}\boldsymbol{x} + \boldsymbol{c})]$ can approximate the following vector operations for arbitrary input $\boldsymbol{x}$:

- Simulate a 2-layer MLP with ReLU activation: $\boldsymbol{W}_2 \mathrm{ReLU}(\boldsymbol{W}_1'\boldsymbol{x} + \boldsymbol{b}_1') + \boldsymbol{b}_2'$
- Simulate any linear operation $\boldsymbol{W}\boldsymbol{x}$
- Simulate element-wise multiplication: $\boldsymbol{x}_1 \otimes \boldsymbol{x}_2$

**Lemma C.1** (Simulating a 2-Layer ReLU MLP with ReGLU Activation). *A 2-layer MLP with ReGLU activation function can simulate any 2-layer MLP with ReLU activation function.*

*Proof.* Let the ReLU MLP be defined as:

$$g(\boldsymbol{x}) = \boldsymbol{W}_2' \mathrm{ReLU}(\boldsymbol{W}_1'\boldsymbol{x} + \boldsymbol{b}_1') + \boldsymbol{b}_2'.$$

Set the weights and biases of the ReGLU MLP as follows:

$$\boldsymbol{W}_1 = \boldsymbol{0}, \quad \boldsymbol{b}_1 = \boldsymbol{1},$$
$$\boldsymbol{V} = \boldsymbol{W}_1', \quad \boldsymbol{b}_2 = \boldsymbol{b}_1',$$
$$\boldsymbol{W}_2 = \boldsymbol{W}_2', \quad \boldsymbol{b} = \boldsymbol{b}_2'.$$

Then, the ReGLU MLP computes:

$$f(\boldsymbol{x}) = \boldsymbol{W}_2'[(\boldsymbol{0} \cdot \boldsymbol{x} + \boldsymbol{1}) \otimes \mathrm{ReLU}(\boldsymbol{W}_1'\boldsymbol{x} + \boldsymbol{b}_1')] + \boldsymbol{b}_2'.$$

Simplifying:

$$f(\boldsymbol{x}) = \boldsymbol{W}_2'[\boldsymbol{1} \otimes \mathrm{ReLU}(\boldsymbol{W}_1'\boldsymbol{x} + \boldsymbol{b}_1')] + \boldsymbol{b}_2' = \boldsymbol{W}_2' \mathrm{ReLU}(\boldsymbol{W}_1'\boldsymbol{x} + \boldsymbol{b}_1') + \boldsymbol{b}_2' = g(\boldsymbol{x}).$$

Thus, the ReGLU MLP computes the same function as the ReLU MLP. $\square$

**Lemma C.2** (Simulating Linear Operations with ReGLU MLP). *A 2-layer MLP with ReGLU activation can compute any linear operation $f(\boldsymbol{x}) = \boldsymbol{W}\boldsymbol{x} + \boldsymbol{b}$.*

*Proof.* To compute a linear function using the ReGLU MLP, we can set the activation to act as a scalar multiplier of one. Set the weights and biases as:

$$\boldsymbol{W}_1 = \boldsymbol{W}, \quad \boldsymbol{b}_1 = \boldsymbol{b},$$
$$\boldsymbol{V} = \boldsymbol{0}, \quad \boldsymbol{b}_2 = \boldsymbol{1},$$
$$\boldsymbol{W}_2 = \boldsymbol{I}, \quad \boldsymbol{b} = \boldsymbol{0}.$$

Here, $\boldsymbol{I}$ is the identity matrix.

Since $\boldsymbol{V}\boldsymbol{x} + \boldsymbol{b}_2 = \boldsymbol{b}_2 = \boldsymbol{1}$, we have:

$$\mathrm{ReLU}(\boldsymbol{V}\boldsymbol{x} + \boldsymbol{b}_2) = \mathrm{ReLU}(\boldsymbol{1}) = \boldsymbol{1}.$$

Then, the ReGLU MLP computes:

$$f(\boldsymbol{x}) = \boldsymbol{I}\left[(\boldsymbol{W}\boldsymbol{x} + \boldsymbol{b}) \otimes \boldsymbol{1}\right] = \boldsymbol{W}\boldsymbol{x} + \boldsymbol{b}.$$

Thus, any linear operation can be represented by appropriately setting $\boldsymbol{W}_1$, $\boldsymbol{b}_1$, and $\boldsymbol{W}_2$. $\qquad\square$

**Lemma C.3** (Element-wise Multiplication via ReGLU MLP). *A 2-layer MLP with ReGLU activation can compute the element-wise multiplication of two input vectors $\boldsymbol{x}_1$ and $\boldsymbol{x}_2$, that is,*

$$f(\boldsymbol{x}) = \boldsymbol{x}_1 \otimes \boldsymbol{x}_2,$$

*where $\boldsymbol{x} = [\boldsymbol{x}_1; \boldsymbol{x}_2]$ denotes the concatenation of $\boldsymbol{x}_1$ and $\boldsymbol{x}_2$.*

*Proof.* Let $\boldsymbol{x} = [\boldsymbol{x}_1; \boldsymbol{x}_2] \in \mathbb{R}^{2n}$, where $\boldsymbol{x}_1, \boldsymbol{x}_2 \in \mathbb{R}^n$.

Set the weights and biases:

$$\boldsymbol{W}_1 = \begin{bmatrix} \boldsymbol{I}_n \\ \boldsymbol{I}_n \end{bmatrix}, \qquad \boldsymbol{b}_1 = \boldsymbol{0}_{2n},$$
$$\boldsymbol{V} = \begin{bmatrix} \boldsymbol{I}_n \\ -\boldsymbol{I}_n \end{bmatrix}, \qquad \boldsymbol{b}_2 = \boldsymbol{0}_{2n},$$
$$\boldsymbol{W}_2 = \begin{bmatrix} \boldsymbol{I}_n & -\boldsymbol{I}_n \end{bmatrix}, \quad \boldsymbol{b} = \boldsymbol{0}_n.$$

Compute:

$$\boldsymbol{W}_1\boldsymbol{x} + \boldsymbol{b}_1 = \begin{bmatrix} \boldsymbol{x}_1 \\ \boldsymbol{x}_1 \end{bmatrix},$$
$$\boldsymbol{V}\boldsymbol{x} + \boldsymbol{b}_2 = \begin{bmatrix} \boldsymbol{x}_2 \\ -\boldsymbol{x}_2 \end{bmatrix},$$
$$\mathrm{ReLU}(\boldsymbol{V}\boldsymbol{x} + \boldsymbol{b}_2) = \begin{bmatrix} \mathrm{ReLU}(\boldsymbol{x}_2) \\ \mathrm{ReLU}(-\boldsymbol{x}_2) \end{bmatrix}.$$

The element-wise product:

$$(\boldsymbol{W}_1\boldsymbol{x} + \boldsymbol{b}_1) \otimes \mathrm{ReLU}(\boldsymbol{V}\boldsymbol{x} + \boldsymbol{b}_2) = \begin{bmatrix} \boldsymbol{x}_1 \otimes \mathrm{ReLU}(\boldsymbol{x}_2) \\ \boldsymbol{x}_1 \otimes \mathrm{ReLU}(-\boldsymbol{x}_2) \end{bmatrix}.$$

Compute the output:

$$\begin{aligned}
f(\boldsymbol{x}) &= \boldsymbol{W}_2\left[(\boldsymbol{W}_1\boldsymbol{x} + \boldsymbol{b}_1) \otimes \mathrm{ReLU}(\boldsymbol{V}\boldsymbol{x} + \boldsymbol{b}_2)\right] + \boldsymbol{b} \\
&= \boldsymbol{x}_1 \otimes \mathrm{ReLU}(\boldsymbol{x}_2) - \boldsymbol{x}_1 \otimes \mathrm{ReLU}(-\boldsymbol{x}_2) \\
&= \boldsymbol{x}_1 \otimes (\mathrm{ReLU}(\boldsymbol{x}_2) - \mathrm{ReLU}(-\boldsymbol{x}_2)) \\
&= \boldsymbol{x}_1 \otimes \boldsymbol{x}_2.
\end{aligned}$$

Thus, the ReGLU MLP computes $f(\boldsymbol{x}) = \boldsymbol{x}_1 \otimes \boldsymbol{x}_2$ without restrictions on $\boldsymbol{x}_2$. $\qquad\square$

### C.2.2 CAPABILITIES OF THE SELF-ATTENTION LAYER

In this subsection, we provide 2 core lemmas on the capabilities of the self-attention layer from Feng et al. (2023).

Let $n \in \mathbb{N}$ be an integer and let $\boldsymbol{x}_1, \boldsymbol{x}_2, \cdots, \boldsymbol{x}_n$ be a sequence of vectors where $\boldsymbol{x}_i = (\tilde{\boldsymbol{x}}_i, r_i, 1) \in [-M, M]^{d+2}$, $\tilde{\boldsymbol{x}}_i \in \mathbb{R}^d$, $r_i \in \mathbb{R}$, and $M$ is a large constant. Let $\boldsymbol{K}, \boldsymbol{Q}, \boldsymbol{V} \in \mathbb{R}^{d' \times (d+2)}$ be any matrices with $\|\boldsymbol{V}\|_\infty \leq 1$, and let $0 < \rho, \delta < M$ be any real numbers. Denote $\boldsymbol{q}_i = \boldsymbol{Q}\boldsymbol{x}_i$, $\boldsymbol{k}_j = \boldsymbol{K}\boldsymbol{x}_j$, $\boldsymbol{v}_j = \boldsymbol{V}\boldsymbol{x}_j$, and define the *matching set* $\mathcal{S}_i = \{j \leq i : |\boldsymbol{q}_i \cdot \boldsymbol{k}_j| \leq \rho\}$. Equipped with these notations, we define two basic operations as follows:

- COPY: The output is a sequence of vectors $\boldsymbol{u}_1, \cdots, \boldsymbol{u}_n$ with $\boldsymbol{u}_i = \boldsymbol{v}_{\mathrm{pos}(i)}$, where $\mathrm{pos}(i) = \operatorname{argmax}_{j \in \mathcal{S}_i} r_j$.
- MEAN: The output is a sequence of vectors $\boldsymbol{u}_1, \cdots, \boldsymbol{u}_n$ with $\boldsymbol{u}_i = \operatorname{mean}_{j \in \mathcal{S}_i} \boldsymbol{v}_j$.

**Assumption C.4.** [Assumption C.6 from Feng et al. (2023)] The matrices $\boldsymbol{Q}, \boldsymbol{K}, \boldsymbol{V}$ and scalars $\rho, \delta$ satisfy that for all considered sequences $\boldsymbol{x}_1, \boldsymbol{x}_2, \cdots, \boldsymbol{x}_n$, the following hold:

- For any $i, j \in [n]$, either $|\boldsymbol{q}_i \cdot \boldsymbol{k}_j| \leq \rho$ or $\boldsymbol{q}_i \cdot \boldsymbol{k}_j \leq -\delta$.

- For any $i, j \in [n]$, either $i = j$ or $|r_i - r_j| \geq \delta$.

Assumption C.4 says that there are sufficient gaps between the attended position (e.g., $\mathrm{pos}(i)$) and other positions. The two lemmas below show that the attention layer with casual mask can implement both COPY operation and MEAN operation efficiently.

**Lemma C.5** (Lemma C.7 from Feng et al. (2023)). *Assume Assumption C.4 holds with $\rho \leq \frac{\delta^2}{8M}$. For any $\epsilon > 0$, there exists an attention layer with embedding size $O(d)$ and one causal attention head that can approximate the COPY operation defined above. Formally, for any considered sequence of vectors $\boldsymbol{x}_1, \boldsymbol{x}_2, \ldots, \boldsymbol{x}_n$, denote the corresponding attention output as $\boldsymbol{o}_1, \boldsymbol{o}_2, \ldots, \boldsymbol{o}_n$. Then, we have $\|\boldsymbol{o}_i - \boldsymbol{u}_i\|_\infty \leq \epsilon$ for all $i \in [n]$ with $\mathcal{S}_i \neq \emptyset$. Moreover, the $\ell_\infty$ norm of attention parameters is bounded by $O(\mathrm{poly}(M, 1/\delta, \log(n), \log(1/\epsilon)))$.*

**Lemma C.6** (Lemma C.8 from Feng et al. (2023)). *Assume Assumption C.4 holds with $\rho \leq \frac{\delta\epsilon}{16M \ln(\frac{4Mn}{\epsilon})}$. For any $0 < \epsilon \leq M$, there exists an attention layer with embedding size $O(d)$ and one causal attention head that can approximate the MEAN operation defined above. Formally, for any considered sequence of vectors $\boldsymbol{x}_1, \boldsymbol{x}_2, \ldots, \boldsymbol{x}_n$, denote the attention output as $\boldsymbol{o}_1, \boldsymbol{o}_2, \ldots, \boldsymbol{o}_n$. Then, we have $\|\boldsymbol{o}_i - \boldsymbol{u}_i\|_\infty \leq \epsilon$ for all $i \in [n]$ with $\mathcal{S}_i \neq \emptyset$. Moreover, the $\ell_\infty$ norm of attention parameters is bounded by $O(\mathrm{poly}(M, 1/\delta, \log(n), \log(1/\epsilon)))$.*

### C.3 THEORETICAL CONSTRUCTION

*Preprint Note: We're in the process of reformatting the construction and proof for better organization*

**Notations**

- $p$ denotes the number of variables
- $t_i$ denotes the token at position $i$
- $T_{vars}$ denotes the set of tokens that denote variables and their negations. i.e. '1', '2', ..., 'n', '-1', '-2', ..., '-n'
- $b$ denotes boolean variables

*Proof.* We first describe the encoding format of the formulas and the solution trace format before going into the details of model construction.

**Input Format.** We consider 3-CNF-SAT formulas in the DIMACS representation, with an initial [BOS] token and an ending [SEP] token. Each variable $x_i$ for $i \in [n]$ has 2 associated tokens: i and -i (e.g., 1 and -1), where the positive token indicates that the $i$-th variable appears in the clause while the negative token indicates that the negation of the $i$-th variable appears in the clause. Clauses are separated using the 0 token. For example, the formula

$$(\neg x_2 \vee \neg x_4 \vee \neg x_1) \wedge (x_3 \vee x_4 \vee \neg x_1) \wedge (\neg x_1 \vee \neg x_3 \vee \neg x_2)$$
$$\wedge (x_1 \vee \neg x_2 \vee \neg x_4) \wedge (\neg x_4 \vee x_2 \vee x_1) \wedge (x_1 \vee \neg x_2 \vee x_4)$$

would be represented as:

```
[BOS] -2 -4 -1 0 3 4 -1 0 -1 -3 -2 0 1 -2 -4 0 -4 2 1 0 1 -2 4 0
                              [SEP]
```

**Solution Trace Format.**   The trace keeps track of the order of the assignments made and whether each assignment is a decision (assumption) or a unit propagation (deduction). Literals with a preceding D token are decision literals while other literals are from unit propagation. When the model encounters a conflict between the current assignment and the formula, it performs a backtrack operation denoted by `[BT]` and performs another attempt with the last decision literal negated. In particular, compared to Figure 1, we used D to abbreviate `Assume` and use `[BT]` to abbreviate `Backtrack`

As an example, the solution trace for the above SAT formula would be:
```
[SEP] D 2 D 1 -4 3 [BT] D 2 D -1 -4 [BT] -2 D 3 D 4 -1 SAT
```

**Embedding Layer.**   Our token set consists of one token for each variable and its negation, the separator token `0`, and a special token D to denote where decisions are made. The positional encoding occupies a single dimension and contains the numerical value of the position of the token in the string. (i.e. there exists a dimension $pos$ such that the position embedding of position $i$ is $i \cdot \mathbf{e}_{pos}$)

**Layer 1.**   The first layer prepares for finding the nearest separator token and $D$ token. Let $i$ denote the position index of tokens:

1. Compute $i_{\text{sep}}$ where $i_{\text{sep}} = i$ if the corresponding token $t_i \in \{\text{`0'}, \text{`[SEP]'}, \text{`[BT]'}\}$ and $i_{\text{sep}} = 0$ otherwise

2. Similarly, compute $i_{\text{D}}$ where $i_{\text{D}} = i$ if the corresponding token $t_i = \text{D}$ and $i_{\text{sep}} = 0$ otherwise.

3. Compute $(i-1)^2$, $i^2$ for index equality comparison

The first 2 operations can both be computed using a single MLP layer that multiplies between $i$ from the positional encoding using Lemma C.3. Similarly, the 3rd operation is a multiplication operation that can be performed with Lemma C.3.

**Layer 2.**   This layer uses 2 heads to perform the following tasks:

1. Copy the index and type of the last separator token and stores

$$p_i^{sep\prime} = \max\{j : j \leq i, t_j \in \{\text{`0'}, \text{`[SEP]'}, \text{`[BT]'}\}\}$$
$$b_0 = (t_j = \text{`0'})$$
$$b_{\text{[SEP]}} = (t_j = \text{`[SEP]'})$$
$$b_{\text{[BT]}} = (t_j = \text{`[BT]'})$$

for $j = p_i^{sep\prime}$

2. (Backtrack) Compute the position of the nearest D token $p_i^{\text{D}} = \max\{j : j \leq i, t_j = \text{`D'}\}$

3. Compute $(p_i^{sep\prime})^2$ for index operation

Task 1 can be achieved via the COPY operation from Lemma C.5 with $\boldsymbol{q}_i = 1$, $\boldsymbol{k}_i = i_{\text{sep}}$, $\boldsymbol{v}_j = (j, \mathbb{I}[t_j = \text{`0'}], \mathbb{I}[t_j = \text{`[SEP]'}], \mathbb{I}[t_j = \text{`[UP]'}], \mathbb{I}[t_j = \text{`[BackTrack]'}])$.

Task 2 is highly similar to task 1 and can be achieved using COPY with $\boldsymbol{q}_i = 1$, $\boldsymbol{k}_i = i_{\text{D}}$, $\boldsymbol{v}_j = (j)$

Task 3 is a multiplication operation that can be performed using Lemma C.3.

**Layer 3** This layer uses 1 head to copy the several values from the previous token to the current token. Specifically, this layer computes:

1. The position of the *previous* separator token, not including the current position:
$$p_i^{sep} = \max\{j : j < i, t_j \in \{\text{`0'}, \text{`[SEP]'}, \text{`[UP]'}, \text{`[BackTrack]'}\}\}$$

2. Dermine if the previous token is D: $b_{decision} = (t_{i-1} = \text{`D'})$ i.e., whether the current token is a decision variable

3. (Induction) Compute the offset of the current token to the previous separator token $d_i^{sep} = i - p_i^{sep\prime}$

4. Compute $(p_i^{sep})^2$, for equality comparison at the next layer.

Task 1 and 2 is done by copying $p_i^{sep\prime}$ and $\mathbb{I}[t_i = \text{`D'}]$ from the previous token. Specifically, we use the COPY operation from Lemma C.5 with $\boldsymbol{q}_i = ((i-1)^2, i-1, 1)$ and $\boldsymbol{k}_j = (-1, 2j, -j^2)$ which determines $i - 1 = j$ via $-((i-1) - j)^2 = 0$ and $\boldsymbol{v}_j = (p_i^{sep\prime}, \mathbb{I}[t_i = \text{`D'}])$. Task 4 is a local multiplication operation that can be implemented via Lemma C.3.

**Layer 4.** This layer uses 2 heads to perform the following tasks:

1. Compute the sum of all variable token embeddings after the previous separator to encode a vector representation of assignments and clauses at their following separator token.
$$\mathbf{r}_i = \sum_{j > p_i^{sep}, t_j \in T_{vars}} \mathbf{e}_{id(t_j)} = \sum_{p_j^{sep} = p_i^{sep}, t_j \in T_{vars}} \mathbf{e}_{id(t_j)}$$

2. (Induction) Compute the position of the second-to-last separator $p_i^{sep-} = \max\{j : j < p_i^{sep}, t_j \in \{\text{`0'}, \text{`[SEP]'}, \text{`[UP]'}, \text{`[BackTrack]'}\}\} = p_{p_i^{sep}}^{sep\prime}$, and the corresponding current position in the previous state $p_i^- = p_i^{sep-} + d_i^{sep}$. As a special case for the first state, we also add 4 to $p_i^-$ if $b_{\text{[SEP]}}$ is true, i.e. $p_i^- = p_i^{sep-} + d_i^{sep} + 4 \cdot b_{\text{[SEP]}}$. The additional 4 is the number of variables per clause + 1 to ensure that we don't consider the last clause as an assignment.

3. (Backtrack) Compute the position of the nearest D token to the last separator token $p_i^{D-} = p_{p_i^{sep}}^{D}$.

4. Compute $b_{exceed} = (p_i^- > p_i^{D-} + 1)$, this denotes whether we're beyond the last decision of the previous state.

5. Compare $(p_i^{D-} \leq p_i^-)$ for $b_{\text{BT\_finished}}$ at the next layer.

6. Compare if $p_i^{D-} = p_i^-$ for the $b_{backtrack}$ operator.

7. Compute $b_{copy}' = (p_i^- < p_i^{sep\prime} - 1)$

Task 1 is achieved using a MEAN operation with $\boldsymbol{q}_i = ((p_i^{sep})^2, p_i^{sep}, 1)$, $\boldsymbol{k}_j = ((-1, 2p_j^{sep}, -(p_j^{sep})^2)$, $\boldsymbol{v}_j = \mathbf{e}_{id(t_j)}$ for $t_j \in T_{vars}$. This attention operations results in $\frac{\mathbf{r}_i}{i - p_i^{sep}}$ The MLP layer then uses Lemma C.3 to multiply the mean result by $i - p_i^{sep}$ to obtain the $\mathbf{r}_i$.

Task 2 is achieved using the COPY operation with $\boldsymbol{q}_i = ((p_i^{sep})^2, p_i^{sep}, 1)$, $\boldsymbol{k}_j = (-1, 2j, -j^2)$ and $\boldsymbol{v}_j = p_i^{sep\prime}$. The MLP layer then performs the addition operation the computes $p_i^-$ by Lemma C.2

Similarly, Task 3 is achieved using the COPY operation with $\boldsymbol{q}_i = ((p_i^{sep})^2, p_i^{sep}, 1)$, $\boldsymbol{k}_j = (-1, 2j, -j^2)$ and $\boldsymbol{v}_j = p_i^{\mathbb{D}}$.

**Layer 5.** The third layer uses 5 heads to perform the following tasks:

1. Determine whether the current assignment $\mathbf{r}_i$ satisfies the formula. $b_{sat}$

2. Determine whether the current assignment $\mathbf{r}_i$ results in a contradiction with a clause of the formula $b_{cont}$

3. Find clauses with at least 2 False literals and sum up the unassigned literals in these clauses. This would result in all the variables that can be currently determined via unit propagation. $\mathbf{e}_{UP}$

4. Compute $b_{final} = b_{exceed} \wedge b_{decision}$

5. Compare $b_{no\_decision} = (p_i^{\mathbb{D}} \leq p_i^{sep})$, which denotes whether the current state contains *no* decision variables

6. Compute $b_{\text{BT\_finished}} = (p_i^{\mathbb{D}^-} \leq p_i^-) \wedge b_{\texttt{[BackTrack]}}$

7. Compare $p_i^-$ with $p_i^{\mathbb{D}^-} - 1$ by storing $p_i^- \leq p_i^{\mathbb{D}^-} - 1$ and $p_i^- \geq p_i^{\mathbb{D}^-} - 1$ (to check for equality at the next layer)

8. Compare $b_{\text{backtrack}} = (p_i^- = p_i^{\mathbb{D}^-} - 1)$

To describe the operations performed in this layer, we interpret the $\mathbf{r}_i$ vectors computed in the previous layer as a $2n$-dimensional *binary encoding* of the clause/assignment preceding token $i$. The value at dimension $2j - 1$ is 1 if the clause/assignment contains variable $j$ (1-indexed) in positive polarity and the value at dimension $2j$ is one iff the clause/assignment contains variable $j$ is in negative polarity. For example, the clause `1 -2 4` is represented as the binary vector $\mathbf{r} = [1, 0, 0, 1, 0, 0, 1, 0]$ when the number of variables is $n = 4$. The $\mathbf{r}$ representation for each clause is at the '0' separator following the clause in the input format.

We now define the linear transformation $T[\mathbf{v}_{true}, \mathbf{v}_{false}, \mathbf{v}_{none}] \in \mathbb{R}^{2n \times 2n}$ where $\mathbf{v}_{true}, \mathbf{v}_{false}, \mathbf{v}_{none} \in \{(0,0), (0,1), (1,0), (1,1)\}$. The transformation takes every pair of values in $\mathbf{r}$ corresponding to each variable, determines whether the variable is true, false, or none-existent in the clause/assignment represented by $\mathbf{r}$, and replaces each pair with the corresponding $\mathbf{v}_{true}, \mathbf{v}_{false}, \mathbf{v}_{none}$ value.

For example, when $\mathbf{r} = [1, 0, 0, 1, 0, 0, 1, 0]$, applying $T[(1,1), (1,0), (0,1)]$ will result in $[1, 1, 1, 0, 0, 1, 1, 1]$. Also, $T[(1,0), (0,1), (0,0)]$ is equivalent to the identity operation. Intuitively, the transformation changes 2-element binary vectors representing true, false, and non-existence within the clause/assignment. The transformation is used to construct query and key matrices to satisfy the desired properties of the assignment-clause dot product.

**Parallel Deduction over Clauses** Task 1 (checking satisfiability) is achieved via an MEAN Lemma C.6 with $\mathbf{q}_i = (\mathbf{r}_i, 1)$ and $\mathbf{k}_j = M(-\mathbf{r}_j, c_j^{(1)})$ and $v_j = \mathbf{1}[t_j = \text{`[BOS]'}]$, where

$$c_j^{(1)} = \begin{cases} 0 & t_j = \text{`0'}, \\ -0.5 & t_j = \text{`[BOS]'} \\ -M & otherwise \end{cases}$$

and $M$ is a sufficiently large constant to approximate hard-max with the softmax operation.

Correctness: Consider the case where $\mathbf{r}_i$ denotes the *binary encoding* of the current assignment and $\mathbf{r}_j$ denotes the *binary encoding* of a clause at a '0' separator position. Then $\mathbf{r}_i \cdot \mathbf{r}_j$ denotes the number of common literals in the assignment and the clause, i.e. how many literals in the clause are True according to the assignment $\mathbf{r}_i$. Therefore, the clause is satisfied by the assignment ending at position $i$ as long as $\mathbf{r}_i \cdot \mathbf{r}_j \geq 1$. Since we only consider the $\mathbf{r}_j$ values at the '0' separators as the *binary encoding* of the clause, all these positions have $c_j^{(1)} = 0$. Therefore, $\mathbf{q}_i \cdot \mathbf{k}_j = -M\mathbf{r}_i \cdot \mathbf{r}_j$, which is

0 for non-satisfied clauses and $< -M$ for satisfied clauses. Also notice that since $c_j^{(1)} = -0.5$ for $j = 1$ (i.e. the first [BOS] token), so $\mathbf{q}_i \cdot \mathbf{k}_1 = -\frac{M}{2}$. If the formula is satisfied, all clauses must be satisfied, and each clause must have attention score $\mathbf{q}_i \cdot \mathbf{k}_j < -M$ while the [BOS] token has attention score $-\frac{M}{2}$. For sufficiently large $M$, we can view the softmax operation as selecting the value vector of the largest attention score item, which is [BOS]. Since $v_1 = \mathbf{1}[t_1 = `[BOS]' = 1$, the result of the attention head will be 1. Conversely, if at least one clause is not satisfied, then $\mathbf{q}_i \cdot \mathbf{k}_j = 0$ for that particular clause. As such, the [BOS] token will not be selected and the result of the attention operation will be 0.

Similarly, task 2 (Detecting Conflict) is also achieved via MEAN(Lemma C.6) with $\mathbf{q}_i = (T[(1, 0), (0, 1), (1, 1)]\mathbf{r}_i, 1)$, $\mathbf{k}_j = M(-\mathbf{r}_j, c_j^{(1)})$, $v_j = 1 - \mathbf{1}[t_j = `[BOS]']$, where the definition of $M$ and $c_j^{(1)}$ is the same as Task 1.

For task 3 (unit propagation), apply MEAN (Lemma C.6) with $\mathbf{q}_i = (T[(0, 1), (1, 0), (0, 0)]\mathbf{r}_i, 1)$, $\mathbf{k}_j = M(\mathbf{r}_j, c_j^{(2)})$, $\mathbf{v}_j = c\mathbf{r}_j$ where

$$c_j^{(2)} = \begin{cases} 0 & t_j = `0', \\ 1.5 & t_j = `[BOS]' \\ -M & otherwise \end{cases}$$

Let the attention result be $\mathbf{o}_{UP}$. The MLP layer then computes $\mathbf{e}_{UP} = ReLU(\mathbf{o}_{UP}) - ReLU(\mathbf{o}_{UP} - 1) - T[(1, 1), (1, 1), (0, 0)]\mathbf{r}_i$ via Lemma C.1.

Correctness: Here we show that, if the assignment at position $i$ does not make the formula unsatisfied, then the resulting vector is approximately a binary encoding of all literals that can be unit-propagated. Consider again the case where $\mathbf{r}_i$ denotes the *binary encoding* of the current assignment and $\mathbf{r}_j$ denotes the *binary encoding* of a clause at a `0' separator position. Here $\mathbf{q}_i \cdot \mathbf{k}_j$ denotes $M$ times the number of false literals in clause $j$ according to the current assignment $i$. Since each clause has three variables, if a clause has three false assignments, then the formula is unsatisfied by the assignment and thus requires no further unit propagation. Therefore, we consider the case where each clause has at most 2 opposing assignments.

If there are no clauses with 2 opposing assignments, then all clause attention logits $\mathbf{q}_i \cdot \mathbf{k}_j$ will be at most $M$, while the attention logit to the [BOS] token will be $c_1^{(2)} = 1.5M$. Since $M$ is a large number, most attention weights will be assigned to [BOS] after the softmax operation and result in a zero embedding vector.

If at least one clause has 2 opposing assignments, all these clauses will have attention logits $\mathbf{q}_i \cdot \mathbf{k} \approx 2M$. Therefore, the attention value will be evenly distributed on all clauses with 2 opposing assignments. The resulting attention output $\mathbf{o}_{[UP]}$ will be the *average of embedding* of all clauses with 2 opposing assignments, multiplied by $c$ since $\mathbf{v}_j = c \cdot \mathbf{r}_j$. Since there are at most $c$ clauses, the number of attended clauses is at most $c$, and the divisor when computing the average is at most $c$. Therefore, the resulting $\mathbf{o}_{[UP]}$ will be a embedding vector where every literal that appeared in at least one clause with 2 false literals have their corresponding position assigned to a $\geq 1$ value.

**Layer 6**  This layer does the remaining boolean operators required for the output. In particular,

- $b_{unsat} = b_{no\_decision} \wedge b_{cont}$

- $b_{[BT]} = b_{cont} \wedge \neg(t_i = [BT])$

- Compute a vector that is equal to $b_{backtrack} \cdot \mathbf{e}_{BT}$, which is equal to $\mathbf{e}_{BT}$ if $b_{backtrack}$ is True and $\mathbf{0}$ otherwise. This is to allow the operation at the output layer for backtracking

Note that $\wedge$ can be implemented as a single ReLU operation for tasks 1 and 2 that can be implemented with Lemma C.1, and task 3 is a multiplication operation implemented with Lemma C.3

**Layer 7**  This layer performs a single operation with the MLP layer: Compute $b_{copy} \cdot e_{copy}$, which gates whether $e_{copy}$ should be predicted based on $b_{copy}$. This enables condition 5 at the output layer.

**Output Projection**    The final layer is responsible for producing the output of the model based on the computed output of the pervious layers. We constructed prioritized conditional outputs, where the model outputs the token according to the first satisfied conditional in the order below:

1. If $b_{sat}$ output `SAT`

2. If $b_{cont} \wedge b_{no\_decision}$ output `UNSAT`

3. If $b_{cont} \wedge \neg(t_i = \texttt{[BackTrack]})$ output '`[BackTrack]`'

4. (BackTrack) If $b_{backtrack}$, output the negation of the token from position $p_i^{\texttt{D}-} + 1$

5. (Induction) If $b_{copy}$, copy token from position $p_i^- + 1$ as output ($e_{copy}$)

6. output a unit propagation variable, if any.

7. output `D` if the current token is not `D`

8. output a unassigned variable

For the output layer, we use $l_{\texttt{[TOKEN]}}$ to denote the output logit of `[TOKEN]`. Since the final output of the model is the token with the highest logit, we can implement output priority by assigning outputs of higher priority rules with higher logits than lower priority rules. Specifically, we compute the output logits vector using the output layer linear transformation as:

$$2^7 \cdot b_{sat} \cdot \mathbf{e}_{\texttt{SAT}} + 2^6 \cdot b_{cont} \cdot \mathbf{e}_{\texttt{[BackTrack]}} + 2^5 \cdot b_{unsat} \cdot \mathbf{e}_{\texttt{UNSAT}}$$

$$+2^4 \cdot b_{backtrack} \cdot \mathbf{e}_{BT} + 2^3 \cdot b_{copy} \cdot \mathbf{e}_{copy} + 2^2 \cdot \mathbf{e}_{\texttt{UnitPropVar}} + 2^1 \cdot (1 - \mathbf{1}[t_i = \text{`D'}]) \cdot \mathbf{e}_{\texttt{D}} + 2^0 \cdot T[(0,0),(0,0),(1,1)]\mathbf{r}_i$$

$$\square$$

**Proposition C.7.** *There exists a transformer with 7 layers, 5 heads, $O(p)$ embedding dimension, and $O(p^2)$ weights that, on all inputs $\boldsymbol{s} \in \mathrm{DIMACS}(p, c)$, predicts the same token as the output as the above operations. Furthermore, let $l_{ctx} = 4c + p \cdot 2^p$ be the worst-case maximum context length required to complete SAT-solving, then all weights are within $\mathrm{poly}(l_{ctx})$ and can be represented within $O(p + \log c)$ bits.*

We only argue from a high level why this is true due to the complexity of the construction. In the above construction, we demonstrate how each operation can be approximated by a Self-attention or MLP layer. We can set the embedding dimension to the sum of dimensions of all the intermediate values and allocate for every intermediate values a range of dimensions that's equal to the dimension of the variables. All dimensions are initialized to 0 in the positional encoding of the transformer except for the dimensions assigned to the positional index $i$. Similarly, only the dimensions assigned to the one-hot token representation are initialized in the token embeddings. At each layer, the self-attention heads and MLP layers extract the variable values from the residual stream and perform the operations assigned to them at each layer.

The only intermediate values whose dimensions are dependent on $p$ are the vectors for one-hot encodings and storing binary encodings of clauses and assignments. They all have size $2p$. Therefore, the number of total allocated embedding sizes is also $O(p)$.

Furthermore,  shows that all parameter values are polynomial with respect to the context length and the inverse of approximation errors. Note that we need only guarantee the final error is less than 1 to prevent affecting the output token. Furthermore, we can choose all parameter values so that they are multiples of 0.5. As such, all parameters are within $\mathrm{poly}(l_{ctx})$ and can be represented by $O(\log(l_{ctx})) = O(p + \log c)$

## C.4    CORRECTNESS

*Note: This section assumes prior knowledge in propositional logic and SAT solving, including an understanding of the DPLL algorithm. For a brief explanation of the notations in this section, please refer to (Nieuwenhuis et al. (2005)). For more general knowledge, please refer to (Biere et al. (2009)).*

We prove that the above model autoregressive solves 3-SAT$_{p,c}$ by showing that it uses the CoT to simulate the "Abstract DPLL Procedure".

### C.4.1 ABSTRACT DPLL

In this section, we provide a description of abstract DPLL. Since the focus of this paper is not to show the correctness of the DPLL algorithm but rather how our model's CoT is equivalent to it, we only present the main results from Nieuwenhuis et al. (2005) and refer readers to the original work for proof of the theorems.

Let $M$ be an ordered trace of variable assignments with information on whether each assignment is an *decision literal* (i.e. assumption) or an *unit propagation* (i.e., deduction).

For example, the ordered trace $3^d\, 1\, \overline{2}\, 4^d\, 5$ denotes the following sequence of operations:

Assume $x_3 = T \rightarrow$ Deduce $x_1 = T \rightarrow$ Deduce $x_2 = F \rightarrow$ Assume $x_4 = T \rightarrow$ Deduce $x_5 = T$.

Let $F$ denote the a SAT formula in CNF format (which includes 3-SAT), $C$ denote a clause (e.g., $x_1 \vee \neg x_2 \vee x_3$), $l$ denote a single literal (e.g., $\neg x_2$), and $l^d$ denote a decision literal. Let $M \models F$ denote that the assignment in $M$ satisfies the formula $F$.

**Definition C.8** (State in the DPLL Transition System). *A state $S \in \mathbb{S}$ in the DPLL transition system is either:*

- The special states SAT, UNSAT, indicating that the formula satisfiable or unsatisfiable

- A pair $M \parallel F$, where:

  - $F$ is a finite set of clauses $C_1 \wedge C_2 \cdots \wedge C_c$ (a conjunctive normal form (CNF) formula), and
  - $M$ is a sequence of annotated literals $l_1 \circ l_2 \cdots \circ l_i$ for some $i \in [n]$ representing variable assignments, where $\circ$ denotes concatenation. Annotations indicate whether a literal is a decision literal (denoted by $l^{\mathrm{d}}$) or derived through unit propagation.

We denote the empty sequence of literals by $\emptyset$, unit sequences by their only literal, and the concatenation of two sequences by simple juxtaposition. While $M$ is a sequence, it can also be viewed as a set of variable assignments by ignoring annotations and order.

**Definition C.9** (Adapted from Definition 1 of Nieuwenhuis et al. (2005)). *The Basic DPLL system consists of the following transition rules $\mathbb{S} \Longrightarrow \mathbb{S}$:*

UnitPropagate :

$$M \parallel F \wedge (C \vee l) \quad \Longrightarrow \quad M \circ l \parallel F \wedge (C \vee l) \quad \textbf{if} \quad \begin{cases} M \models \neg C, \\ l \text{ is undefined in } M. \end{cases}$$

Decide :

$$M \parallel F \quad \Longrightarrow \quad M \circ l^{\mathrm{d}} \parallel F \quad \textbf{if} \quad \begin{cases} l \text{ or } \neg l \text{ occurs in a clause of } F, \\ l \text{ is undefined in } M. \end{cases}$$

Backjump :

$$M \circ l^{\mathrm{d}} \circ N \parallel F \quad \Longrightarrow \quad M \circ l' \parallel F \quad \textbf{if} \quad \begin{cases} \text{There is some clause } C \vee l' \text{ s.t.} \\ F \models C \vee l', \quad M \models \neg C, \\ l' \text{ is undefined in } M, \\ l' \text{ or } \neg l' \text{ occurs in a clause of } F. \end{cases}$$

Fail :

$$M \parallel F \wedge C \quad \Longrightarrow \quad \text{UNSAT} \quad \textbf{if} \quad \begin{cases} M \models \neg C, \\ M \text{ contains no decision literals.} \end{cases}$$

Success :

$$M \parallel F \quad \Longrightarrow \quad \text{SAT} \quad \textbf{if} \quad M \models F$$

We also use $S \Longrightarrow^* S'$ to denote that there exist $S_1, S_2, \ldots, S_i$ such that $S \Longrightarrow S_1 \Longrightarrow \cdots \Longrightarrow S_i \Longrightarrow S'$. Also $S \Longrightarrow^! S'$ denote that $S \Longrightarrow^* S'$ and $S'$ is a final state (SAT or UNSAT).

*Explanation of the* Backjump *Operation:*

The Backjump operation allows the DPLL algorithm to backtrack to a previous decision and learn a new literal. In particular, $F \models C \vee l'$ means that, for some clause $C$, every assignment that satisfies $F$ must either satisfy $C$ (i.e., contain the negation of each literal in $C$) or contain $l'$ as an assignment. However, if $M \models \neg C$, which means that $M$ conflicts with $C$ and thus contains the negation of each literal in $C$, then if we want some assignment containing $M$ to still satisfy $F$, then the assignment must also include the literal $l'$ as an assignment to ensure that it satisfies $C \vee l'$, a requirement for satisfying $F$.

In our construction, we only consider the narrower set of BackTrack operations that find the last decision and negate it:

**Lemma C.10.** *[Corrollary of Lemma 6 from Nieuwenhuis et al. (2005)] Assume that $\emptyset \parallel F \Longrightarrow^*$ $M \circ l^{\mathrm{d}} \circ N \parallel F$, the* BackTrack *operation:*

$$
M \circ l^{\mathrm{d}} \circ N \parallel F \qquad \Longrightarrow \qquad M \circ \neg l \parallel F \qquad \textit{if} \quad \begin{cases} \textit{There exists clause } C \textit{ in } F \textit{ such that} \\ M \circ l^{\mathrm{d}} \circ N \models \neg C \\ N \textit{ contains no decision literals} \end{cases}
$$

*is always a valid* Backjump *operation in Definition C.9.*

**Definition C.11** (Run of the DPLL Algorithm). A *run* of the DPLL algorithm on formula $F$ is a sequence of states $S_0 \Longrightarrow S_1 \Longrightarrow \cdots \Longrightarrow S_T$ such that:

- $S_0$ is the initial state $\emptyset \parallel F$

- For each $i = 0, 1, \ldots, n-1$, the transition $S_i \Longrightarrow S_{i+1}$ is valid according to the transition rules of the DPLL system in Definition C.9 (e.g., UnitPropagate, Decide, Backjump, or Fail);

- $S_n$ is a final state that is either SAT or UNSAT

Note that the above definition is simply the expansion of $\emptyset \parallel F \Longrightarrow^! S_T$.

The following theorem states that the DPLL procedure always decides the satisfiability of CNF formulas:

**Lemma C.12.** *[Theorem 5 and Theorem 9 Combined from Nieuwenhuis et al. (2005)] The Basic DPLL system provides a decision procedure for the satisfiability of CNF formulas $F$. Specifically:*

1. *$\emptyset \parallel F \Longrightarrow^!$ UNSAT if and only if $F$ is unsatisfiable.*

2. *$\emptyset \parallel F \Longrightarrow^!$ SAT if and only if $F$ is satisfiable.*

3. *There exist no infinite sequences of the form $\emptyset \parallel F \Longrightarrow S_1 \Longrightarrow \cdots$*

### C.4.2 Trace Equivalence and Inductive Proof

We demonstrate that our Transformer in Theorem 4.5 solves SAT by showing that the CoT produced by the Transformer is "trace equivalent" to an abstract DPLL algorithm with some heuristic. We first provide definition of "trace equivalence":

**Definition C.13** (Trace Equivalence of Algorithms). Let $A$ and $B$ be two algorithms. Let $\Sigma_A$ and $\Sigma_B$ be the sets of possible states of $A$ and $B$, respectively. We say that algorithms $A$ and $B$ are *trace equivalent* if there exists a bijective mapping $\phi : \Sigma_A \to \Sigma_B$, independent of the input, such that for every input $s$, the traces produced by $A$ and $B$ satisfy the following:

If the execution of $A$ on input $s$ produces the trace $\mathrm{Tr}_A(s) = [\sigma_1^A, \sigma_2^A, \ldots, \sigma_n^A]$, and the execution of $B$ on the same input $s$ produces the trace $\mathrm{Tr}_B(s) = [\sigma_1^B, \sigma_2^B, \ldots, \sigma_n^B]$, then for all $i \in \{1, 2, \ldots, n\}$,

$$
\sigma_i^B = \phi(\sigma_i^A).
$$

That is, the sequences of states of $A$ and $B$ are in one-to-one correspondence via the fixed mapping $\phi$, and corresponding states are related by this mapping for every input $s$.

We first show how to convert a chain of thought of the model into a state in the abstract DPLL algorithm. Consider the following model input and Chain-of-Thought trace:

```
[BOS] -2 -4 -1 0 3 4 -1 0 -1 -3 -2 0 1 -2 -4 0 -4 2 1 0 1 -2 4 0
[SEP] D 2 D 1 -4 3 [BT] D 2 D -1 -4
```

Recall that `[BT]` denotes backtracking and `D` denotes that the next token is a decision literal.

Note that the prompt input ends at `[SEP]` and the rest is the Chain-of-Though produced by the model.

We want to convert this trace to a state $S = M\|F$ such that $F$ is the CNF formula in the DIAMCS encoding in the prompt input and $M$ is the "assignment trace" at the last attempt (i.e., after the last `[BT]` token.). As such, $M$ correspond to the `D 2 D -1 -4` portion of the trace and thus $M = 2^d \, \overline{1}^d \, \overline{4}$ as described in Appendix C.4.1. We formalize this process as follows:

**Definition C.14** (Translating CoT to Abstract DPLL State). For any number of variables $p \in \mathbb{N}^+$, let $\mathcal{V}$ be the set of tokens:

$$\mathcal{V} = \{\, \texttt{-i}, \texttt{i} \mid i \in [p] \,\} \cup \{\, \texttt{D}, \texttt{[SEP]}, \texttt{[BOS]}, \texttt{[BT]}, \texttt{0}, \texttt{SAT}, \texttt{UNSAT} \,\}.$$

Define a mapping $f_{\mathcal{S}} : \mathcal{V}^* \to \mathcal{S} \cup \{\text{error}\}$ that converts a sequence of tokens $R \in \mathcal{V}^*$ into an abstract DPLL state as follows:

1. **If** $R$ ends with `SAT` or `UNSAT`, **then** set $M_{\mathcal{S}}(R)$ to SAT or UNSAT accordingly.

2. **Else if** $R$ contains exactly one `[SEP]` token, split $R$ at `[SEP]` into $R_{\text{DIMACS}}$ and $R_{\text{Trace}}$.

3. Parse $R_{\text{DIMACS}}$ into a CNF formula $F$, assuming it starts with `[BOS]` and ends with `0`. If parsing fails, set $M_{\mathcal{S}}(R) = \text{fail}$.

4. Initialize an empty sequence $M$ to represent variable assignments and set a flag *isDecision* $\leftarrow$ `False`.

5. Process each token $t$ in $R_{\text{Trace}}$ sequentially:

    - **If** $t = \texttt{D}$, set *isDecision* $\leftarrow$ `True`.
    - **Else if** $t = \texttt{[BT]}$, remove literals from $M$ up to and including the last decision literal (i.e., perform backtracking).
    - **Else if** $t = \texttt{i}$ or $\texttt{-i}$ for some $i \in [n]$:
        - Let $l$ be the literal corresponding to $x_i = T$ if $t = \texttt{i}$, or $x_i = F$ if $t = \texttt{-i}$.
        - **If** $l$ is already assigned in $M$ with a conflicting value, set $M_{\mathcal{S}}(R) = \text{fail}$.
        - **Else**, append $l$ to $M$, annotated as a decision literal if *isDecision* = `True`, or as a unit propagation otherwise.
        - Reset *isDecision* $\leftarrow$ `False`.
    - **Else**, set $M_{\mathcal{S}}(R) = \text{error}$.

6. **Return** the state $M \parallel F$.

7. **If** any of the above steps fail, set $M_{\mathcal{S}}(R) = \text{fail}$.

We now present the inductive lemma:

**Lemma C.15** (Inductive Lemma). *For any $p, c \in \mathbb{N}^+$, for any input $F_{DIMACS} \in \text{DIMACS}(p, c)$ of length $n$, let $F$ be the boolean formula in CNF form encoded in $F_{DIMACS}$. Let $A$ be the model described in section C.3 with parameters $p, c$. Let $(\boldsymbol{s}_{1:n}, \boldsymbol{s}_{1:n+1}, \dots)$ be the trace of $\boldsymbol{s}$ when running the Greedy Decoding Algorithm 1 with model $A$ and input prompt $\boldsymbol{s}_{1:n} = F_{DIMACS}$. For every $i \in \mathbb{N}^+$, if $f_{\mathcal{S}}(\boldsymbol{s}_{1:n+i}) = S$ and $S \notin \{\text{SAT}, \text{UNSAT}, \text{error}\}$, then there exist $j \in \mathbb{N}^+$ and $S' \in \mathbb{S}$ such that $S \implies S'$ and $f_{\mathcal{S}}(\boldsymbol{s}_{1:n+i+j}) = S'$.*

We now show trace equivalence between the model $A$ and some instantiating of the abstract DPLL with a specific heuristic:

**Definition C.16.** For any heuristic $h : \mathbb{S} \to \mathcal{L}$ where $\mathcal{L}$ is the set of literals, let $\text{DPLL}_h$ denote an instantiation of the abstract DPLL algorithm that selects $h(S)$ as the decision literal when performing Decide and only performs the BackTrack operation for Backjump. $h(S)$ is a valid heuristic if $\text{DPLL}_h$ always abides by the Decide transition.

**Lemma C.17.** *(Trace Simulation) There exists a valid heuristic $h : \mathbb{S} \to \mathcal{L}$ for which the Transformer model $A$ is trace equivalent to $\text{DPLL}_h$ on all inputs in $\text{DIMACS}(p, c)$*

*Proof.* We aim to show that there exists a valid heuristic $h : \mathcal{S} \to \mathcal{L}$ such that the Transformer model $A$ is trace equivalent to $\text{DPLL}_h$ on all inputs in $\text{DIMACS}(p, c)$.

Define the heuristic $h$ as follows: For any state $S \in \mathcal{S}$, let $h(S)$ be the literal that the Transformer model $A$ selects as its next decision literal when in state $S$.

Formally, given that the model $A$ outputs tokens corresponding to decisions, unit propagations, backtracks, etc., and that these tokens can be mapped to transitions in the abstract DPLL system via the mapping $M_{\mathcal{S}}$ (as per the *Translating CoT to Abstract DPLL State* definition), we set:

$$h(S) = \begin{cases} \text{the decision literal chosen by } A \text{ in state } S, & \text{if } A \text{ performs a Decide transition,} \\ \text{undefined}, & \text{otherwise.} \end{cases}$$

This heuristic is valid because $A$ always abides by the Decide transition rules, ensuring $h(S)$ selects a literal that occurs in $F$ and is undefined in $M$, satisfying the conditions of a valid heuristic.

Define a mapping $\phi : \Sigma_A \to \Sigma_B$, where $\Sigma_A$ is the set of possible states of model $A$, and $\Sigma_B$ is the set of possible states of $\text{DPLL}_h$, such that for any state $S$ in the execution trace of $A$, $\phi(S) = S$. That is, we identify the states of $A$ with the corresponding states in $\text{DPLL}_h$ by mapping the sequence of assignments and the formula $F$ directly.

**Proof of Trace Equivalence:**

We proceed by induction on the number of steps in the execution trace.

*Base Case ($i = 0$):*

At the beginning, both algorithms start from the initial state with no assignments:

$$\text{For } A : \quad S_0^A = \emptyset \parallel F, \quad \text{and} \quad \text{For } \text{DPLL}_h : \quad S_0^B = \emptyset \parallel F.$$

Clearly, $\phi(S_0^A) = S_0^B$.

*Inductive Step:*

Assume that after $k$ steps, the states correspond via $\phi$:

$$\phi(S_k^A) = S_k^B.$$

We need to show that after the next transition, the states still correspond, i.e., $\phi(S_{k+1}^A) = S_{k+1}^B$.

Suppose the model $A$ applies a UnitPropagate operation, transitioning from state $S_k^A$ to $S_{k+1}^A$ by adding a literal $l$ deduced via unit propagation.

Since unit propagation is deterministic and depends solely on the current assignment $M$ and formula $F$, $\text{DPLL}_h$ will also apply the same UnitPropagate operation, transitioning from $S_k^B$ to $S_{k+1}^B$ by adding the same literal $l$.

Thus, $\phi(S_{k+1}^A) = S_{k+1}^B$.

Suppose the model $A$ applies a Decide operation, transitioning from $S_k^A$ to $S_{k+1}^A$ by adding a decision literal $l = h(S_k^A)$.

By the definition of the heuristic $h$, $\text{DPLL}_h$ also selects $l$ as the decision literal in state $S_k^B$. Both algorithms make the same decision and transition to the same next state.

Therefore, $\phi(S_{k+1}^A) = S_{k+1}^B$.

Suppose the model $A$ applies a Backjump operation, backtracking to a previous state and assigning a new literal.

Since $\text{DPLL}_h$ performs only the $\text{BackTrack}$ operation for $\text{Backjump}$ (as per the definition), and $A$ simulates this operation, both algorithms backtrack in the same manner and update their assignments accordingly.

Thus, $\phi(S^A_{k+1}) = S^B_{k+1}$.

If the model $A$ reaches a terminal state indicating SAT or UNSAT, then so does $\text{DPLL}_h$, since their sequences of transitions have been identical up to this point.

In all cases, the next state of model $A$ corresponds to the next state of $\text{DPLL}_h$ under the mapping $\phi$. Therefore, by induction, the execution traces of $A$ and $\text{DPLL}_h$ are such that for all $i$,

$$\phi(S^A_i) = S^B_i.$$

Since the heuristic $h$ selects the same decision literals as the model $A$, and $A$ always abides by the $\text{Decide}$ transition (as per its design), $h$ is a valid heuristic according to the definition provided.

$\square$

# D  CODE FOR THEORETICAL CONSTUCTION

```python
def nearest_token_id(tok_emb: OneHotTokEmb, vocab: List[str],
                     targets: List[str], indices: Indices=indices):
    # Get the token ids of the target tokens
    target_tok_ids = [vocab.index(target) for target in targets]
    # Get whether the current token is one of the target tokens
    # by summing the one-hot embedding
    target_token_embs = Concat([tok_emb[:, target_tok_id]
                                for target_tok_id in target_tok_ids])
    in_targets = target_token_embs.sum(axis=1)
    # Filter the indices to only include the target tokens
    filtered_index = indices * in_targets
    return filtered_index.max()
```

