# OpenReview forum: "Can Transformers Reason Logically? A Study in SAT Solving"
_ICLR.cc/2025/Conference — Submitted to ICLR 2025_

### Official Review · Reviewer_6FmX · 2024-10-29

**Soundness:** 2
**Presentation:** 2
**Contribution:** 3
**Rating:** 5
**Confidence:** 2

**Summary:**

The paper contributes to research on understanding the reasoning capabilities of transformers when using the chain of thought. Its contributions are threefold:
- It asserts that a decoder-only transformer can, in principle, solve Boolean SAT using the chain-of-thought to reason similar to the DPLL algorithm.
- It introduces PARAT, described as a "compiler that takes procedural specifications as input and outputs a transformer model that implements these specifications."
- It empirically explores whether a transformer can be trained to solve SAT problems.

**Strengths:**

- Investigating the reasoning capabilities of transformers, particularly concerning formal
problems, is highly valuable.
- Boolean satisfiability serves as an appropriate baseline candidate.
- Supporting the initial contribution of theoretical expressiveness with experiments that
provide a practical perspective is a sound approach.

**Weaknesses:**

My main concerns regard the first two contributions. I categorize the weaknesses based on the contributions of this paper and order them from most important to least important in each category.

### Transformers with CoT solve SAT
- The claim of the paper may be perceived as somewhat misleading. While it states,
  "we construct a decoder-only Transformer that can solve SAT," the actual result
  (Theorem 1) is that "for any $p,c \in \mathbb{N}^+$ there exists a decoder-only Transformer
  with [...] that can decide all 3SAT instances of at most $p$ variables and $c$ clauses using CoT."
  This is correctly described as the construction of a decoder-only Transformer for
  a bounded version of 3SAT. Alternatively, Theorem 1 implies that the class of all
  decoder-only Transformers solves the general 3SAT problem: given an arbitrary
  3SAT formula, there exists a decoder-only Transformer that can reason whether it is satisfiable.
  The distinction is significant, as this contribution of the paper is directed towards proving
  that a single decoder-only Transformer exists for solving SAT. In fact this is not implied by Theorem 1.
- Definition 4.3 could benefit from clarification. While the intended meaning might be inferred,
  there remains uncertainty due to the probabilistic nature of the map computed by a decoder-only
  transformer. The pivotal question is ensuring the decision procedure always
  terminates. It appears that a parameter may be absent in this definition, one that would
  guarantee termination after a predetermined length if an end-of-sequence token does not
  appear.
- The intuition behind the Transformer template constructed for Theorem 1 needs improvement.
  While Figure 1 is comprehensive, despite being oddly distant in placement within the paper,
  the explanations from line 277 to 300 are overly abstract. Specifically, lines 288 to 294
  lack clarity, making it difficult for the reader to grasp their meaning. This may confuse
  the reader more than it helps.

### Parat
- The main concern is that the paper struggles to clearly define what PARAT is. The initial description
between lines 85 to 87 reads, "We design PARAT, a compiler of high-level sequence operations in Numpy-like
syntax into Transformer model weights, to empirically validate and analyse theoretical constructions of
Transformer algorithms." However, there is ambiguity surrounding the terms "compiler", "high-level sequence operations", "Transformer model weights", and "Transformer algorithms". Further on line 306, it states, "... a framework for converting theoretical constructions of Transformers into practical models ...". Yet, it remains unclear what is meant by "theoretical constructions of Transformers" and how these relate to "high-level sequence operations". The first detailed exposition of PARAT (commencing with section 5.1) immediately delves into features and operations, leaving a reader, such as myself, completely perplexed as to what it truly encompasses.
- Without addressing the aforementioned ambiguities, it remains unclear why PARAT's contribution is significant for exploring the reasoning capabilities of Transformers, the paper's primary objective. While it appears to be a useful tool, developed with considerable effort, it seems to distract from the main aim of the paper and takes up space that could be used to better elucidate the first contribution.

**Questions:**

Having shared broad and high-level concerns regarding the paper, I encourage the authors to address these directly.
Here is a summary of the key points:
- Do you agree that your theoretical contributions do not demonstrate the existence of a decoder-only Transformer that solves SAT? Do you believe this could be resolved with minor revisions, and if so, how would you approach this?
- How do you ensure the decision procedure in Definition 4.3 always terminates? What considerations
lead to defining the "semantics" of a decoder-only transformer as mapping to a probability
distribution? Is this crucial? If I am not mistaken, this perspective somewhat diverges from research
addressing the expressiveness of transformers, such as the paper "Attention is Turing-complete", which you cite, or studies focusing on the expressiveness of single-pass or encoder-only transformers.
- Can you provide a clear and concise definition of what Parat is?

---

> ### Author Response · Authors · 2024-11-23
> **Thank you for your detailed and insightful reviews and reply to concerns [Part 1]**
>
> Thank you for your careful review of the paper and for pointing out the important distinction between different interpretations of our theorems. We will answer your three questions and address your concerns separately in the following sections.
>
> ## **Q1: Concern Can Be Fixed via a Minor Revision**
>
> We fully agree that our theorem does not demonstrate the existence of a decoder-only Transformer that solves SAT, and our current expression in the abstract is flawed. Our theorem implies that different Transformer models are required for different **input sizes** of 3-SAT instances. The accurate statement would be: “We theoretically construct non-uniform decoder-only Transformers that can solve SAT using backtracking and deduction via Chain-of-Thought (CoT).”
>
> However, after this minor revision, we believe that our claim that “decoder-only Transformers can solve 3-SAT” is well-founded, non-trivial, and follows the expression conventions of seminal works on Transformer expressiveness. In particular, our theorem considers the setting of non-uniform computation, which is significantly more powerful than “given an arbitrary 3-SAT formula, there exists a decoder-only Transformer that can reason whether it is satisfiable.” The non-uniform computation setting is also implicitly assumed throughout the relevant literature.
>
> To support this claim, we will divide our arguments as follows:
>
> ## **1.1 Clarification of Our Theorem as a Non-Uniform Computational Model and Why It Is Non-Trivial**
>
> The dependence of the construction on the number of clauses $c$ demonstrates that a single Transformer model can achieve logical reasoning within bounded input length. However, this capability is already highly non-trivial. To demonstrate this, we first differentiate between different kinds of computation models:
>
> 1.	**Uniform Computation**: There exists a single Transformer model that can solve all 3-SAT instances regardless of input size.
>
> 2.	**Non-Uniform Computation**: Given any input **size** $n$, there exists a Transformer model that can decide all 3-SAT instances with input size up to $n$.
>
> 3.	Given any input 3-SAT **instance** $x$, there exists a Transformer model that can decide $x$.
>
> Option 3 best resembles your description of “given an arbitrary 3-SAT formula, there exists a decoder-only Transformer that can reason whether it is satisfiable.” This interpretation has a trivial proof of existence since we can simply construct a Transformer that outputs the constant string of the correct reasoning for that particular instance by encoding the exact string in the weights of the model. However, our theorem is strictly more powerful than this claim.
>
> Option 1 is the strongest claim but, as we will elaborate in section 1.3, is very likely impossible for Transformer models.
>
> Option 2 is a more realistic setting that is adopted by most works investigating Transformer expressiveness. In our case, the Transformer is only dependent on the maximum number of variables $p$ and the maximum number of clauses $c$, and is not dependent on the actual input instance itself, as in Option 3. Even if we only consider formulas with exactly $p$ variables and $c$ clauses (rather than “at most”), there are more than $p^{3c}$ possible 3-SAT instances that can be solved by a single Transformer model.
>
> To illustrate this quantity and how it compares with the number of parameters in the construction, for the 20 max variable and 88 max clause Transformer we compiled for evaluation, our theorem implies that the single Transformer model is able to correctly reason about the satisfiability of more than $10^{343}$ possible SAT instances. However, the model only contains 5,011,862 parameters, which shows that it is impossible to trivially encode all reasoning paths for all possible samples as parameters of the Transformer. Therefore, our theorem implies that Transformers are capable of performing logical reasoning in the context of 3-SAT solving.
>
> Our theoretical construction also depends on the maximum number of variables $p$. This is due to each variable requiring a unique token in the vocabulary. Therefore, the number of token embeddings and the embedding size have to be increased to account for a larger vocabulary size. Thus, having a different Transformer to account for larger vocabulary sizes is inevitable by design. However, in any given LLM with a fixed vocabulary, the actual number of variables involved in the formulas does not matter for our theoretical construction. Since all Transformers have predefined fixed-size vocabularies, the dependence of our Transformer on the number of variables $p$ is not a significant limitation in reasoning capability. Rather, the dependence on $p$ is due to the token-based processing design of Transformer models and can be ignored with a pre-defined fixed vocabulary.

---

> > ### Author Response · Authors · 2024-11-23
> > **Thank you for your detailed and insightful reviews and reply to concerns [Part 2]**
> >
> > ## **1.2 Non-Uniform Models Implicit in Relevant Works**
> >
> > - In “Chain-of-Thought Empowers Transformers to Solve Inherently Serial Problems” [1], the authors claim that “Transformers with polynomially many intermediate steps are capable of computing all circuits with polynomial size” (Page 2). Since Boolean circuits are fixed in the number of input gates, every input length requires a different Boolean circuit, and different circuits correspond to different Transformer models. Therefore, the non-uniform model of computation is implicitly assumed in this statement.
> > - In “Towards Revealing the Mystery behind Chain of Thought: A Theoretical Perspective” [2], the authors claim that “we then prove by construction that autoregressive Transformers of constant size suffice to solve both tasks” (basic arithmetic/equation tasks). In their theorems, the Transformer model is dependent on both $n$ and $p$ for Arithmetic$(n,p)$, which denotes evaluating arithmetic expressions of length $n$ modulo $p$. This is also non-uniform, as in our work. Similarly, for Equation$(m,p)$, which denotes linear equations with $m$ variables modulo $p$, the Transformer model depends on both $m$ and $p$, which is almost exactly the same dependence as our theorem.
> > - In “Transformers Learn Shortcuts to Automata” [3], the authors claim that “a low-depth Transformer can represent the computations of any finite-state automaton.” However, for a single automaton, a Transformer of $O(\log T)$ depth is required to simulate computation on inputs of size $T$, and thus the Transformer is also different for different input lengths.
> >
> > ## **1.3 Why Non-Uniformity Is Necessary**
> >
> > The main reason for requiring different Transformers for increasing lengths is due to the accumulating error of softmax attention on longer contexts. In particular, many of the operations implemented in our theoretical constructions (as well as in relevant works) require an attention head to attend to a strict subset of previous positions while ignoring the rest. However, due to the properties of softmax, every position must be assigned a non-zero attention value, and when the input lengths increase, the attention values toward unwanted positions grow larger. As such, the parameters for calculating attention need to scale to keep the unwanted attention within a predefined threshold.
> >
> > There are several prior theoretical works that consider Averaging Hard Attention [4][5][6], which circumvents this issue by using an idealized attention operation where all unwanted positions are assumed to have an attention score of exactly zero. If we were to adopt this model, then our corresponding theoretical construction could indeed generalize indefinitely for arbitrary $c$ values. However, this model is not used in practice and is not known to be trainable. Therefore, we consider the more realistic softmax Transformer used in practice and adopt the non-uniform model of computation.
> >
> > [1] Zhiyuan Li, Hong Liu, Denny Zhou, and Tengyu Ma. Chain-of-Thought Empowers Transformers to Solve Inherently Serial Problems
> >
> > [2] Guhao Feng, Bohang Zhang, Yuntian Gu, Haotian Ye, Di He, and Liwei Wang. Towards revealing the mystery behind chain of thought: A theoretical perspective.
> >
> > [3] Bingbin Liu, Jordan T. Ash, Surbhi Goel, Akshay Krishnamurthy, Cyril Zhang, Transformers Learn Shortcuts to Automata
> >
> > [4] William Merrill, Ashish Sabharwal, Noah A. Smith, Saturated Transformers are Constant-Depth Threshold Circuits
> >
> > [5] Jorge P´erez, Pablo Barcel´, Javier Marinkovic, Attention is Turing Complete
> >
> > [6] Pablo Barcelo´, Alexander Kozachinskiy, Anthony Widjaja Lin, Vladimir Podolski, Logical Languages Accepted by Transformer Encoders with Hard Attention.

---

> > > ### Author Response · Authors · 2024-11-23
> > > **Thank you for your detailed and insightful reviews and reply to concerns [Part 3]**
> > >
> > > ## Q2.1 Decision Procedure
> > >
> > > “How do you ensure the decision procedure in Definition 4.3 always terminates? ”
> > >
> > > Thank for for pointing out a flaw in our definitions and we fully agree that it is unclear in our current statement that a decision procedure must terminate. In our intention for Definition 4.3, we use the term “decision procedure” to refer only to procedures that are guanrateed to terminate, in parallel with the definition of deciders for Turing Machines[7]. Therefore, if a procedure does not terminate for some input in $L$, it is not a decision procedure by definition. We have adjusted our paper accordingly (modified Definition 4.2 to explicitly state that the procedure must halt and remove the statement of $\mathcal{A}_{M,\mathcal{E}}$ as a decision procedure in Definition 4.3.) to explicitly state that a decision procedure must always halt and we’re very grateful to the reviewer for the observation.
> > >
> > > However, this does not affect the correctness of our theorems since we were having the above corrected definition in mind when proving and writing the theorem.
> > >
> > > A highly relevant question that we believe you’re referring to is “how do we ensure that our theoretical construction always terminates during autoregressive decoding”. This is done by proving trace equivalence of our theoretical construction with the abstract DPLL algorithm [8], which is proved to always terminate in the referred paper (Theorem 5). In particular, in our proof we define a translation procedure (Definition D.14) from Chain-of-Thought (CoT) prefixes to abstract DPLL states. If the input is a valid 3-SAT formula, for every prefix of the CoT generated by our Theoretical construction, it can always be translated into an abstract DPLL state $s_1$ such that, after generating a bounded number of additional CoT tokens, the new CoT prefix can be translated into an abstract DPLL state $s_2$ such that $s_1\rightarrow s_2$ is a valid abstract DPLL transition. The 2 terminal states of the abstract DPLL algorithm in our formulation (Definition C.9), SAT and UNSAT, correspond to Chain-of-Thought that ends with the tokens SAT and UNSAT respectively. Therefore, since there do not exist any infinite state transitions for abstract DPLL, and the number of CoT tokens required for each abstract DPLL transition is bounded, there must also be no infinite length CoT that can simulate the abstract DPLL algorithm.
> > >
> > > ## Q2.2 Probability Distribution
> > >
> > > Thank you for pointing out the definition of Transformers with probability distribution output as potentially confusing. We mainly formulated Transformers with probability distribution as output to align with the modus operandi of commercial LLMs. However, as you have correctly pointed out, this definition is necessary and is not aligned with definitions in prior works, which can therefore be a source of confusion. We have adjusted the definitions in our paper so that the model directly outputs the next token.
> > >
> > > [7] [https://en.wikipedia.org/wiki/Decider_(Turing_machine)](https://en.wikipedia.org/wiki/Decider_(Turing_machine))
> > >
> > > [8] Robert Nieuwenhuis, Albert Oliveras, and Cesare Tinelli. Abstract dpll and abstract dpll modulo theories

---

> ### Comment · Reviewer_6FmX · 2024-11-23
> **answer to response part 1 + 2**
>
> Thanks for getting back to me, I do my best to address all you responses adequately, since the rebuttal phase is ending soon.
>
> ---
>
> > "We theoretically construct non-uniform decoder-only Transformers that can solve SAT using backtracking and deduction via Chain-of-Thought (CoT)."
>
> What do you mean by "non-uniform" here? Am I right to assume what you want to say is " ... construct a class of decoder-only Transformers that can solve...."?
>
> > "decoder-only Transformers can solve 3-SAT”
>
> Sorry, but I think that this short, informal statement oversells.
>
> > which is significantly more powerful than “given an arbitrary 3-SAT formula, there exists a decoder-only Transformer that can reason whether it is satisfiable.”
>
> Ah, of course, sorry. That was wrongly stated by me, meaning that I switched the quantification. What I meant was "there is a Transformer T that given a 3-SAT formula ...".
>
> I appreciate your extensive explanations regarding the three "levels of computation", but as stated above I never intended to suggest such a weak form of statement to be stronger. Sorry again for the typo!

---

> ### Comment · Reviewer_6FmX · 2024-11-23
> **answer to response 3**
>
> > We have adjusted our paper accordingly (modified Definition 4.2 to explicitly state that the procedure must halt and remove the statement of
>  as a decision procedure in Definition 4.3.) to explicitly state that a decision procedure must always halt and we’re very grateful to the reviewer for the observation.
>
> Am I right that you so far did not revise it?
>
> > This is done by proving trace equivalence of our theoretical construction with the abstract DPLL algorithm [8], which is proved to always terminate in the referred paper (Theorem 5).
>
> While I did not check the details again, this makes sense. Thanks!
>
> > We mainly formulated Transformers with probability distribution as output to align with the modus operandi of commercial LLMs [...] We have adjusted the definitions in our paper so that the model directly outputs the next token.
>
> That is an important difference, you are right! Its probably best to make this clear at some point. Given this LLM view, the "probability distribution view" is clearly the right one. Again, you did not revise anything yet, right?
>
> ---
>
> You did not respond to my question regarding PARAT so far, right?
>
> ---
>
> So far, I believe my initial rating to be correct. I want to highlight that I very much appreciate the theoretical contributions that the paper makes. The answers in the rebuttal make clear that the authors have a very clear understanding of their contributions and the mathematical details are sufficiently supported in the appendix, although I did not check them carefully. However, I still feel the current form of presentation in the paper does not do this justice and strongly obscures the actual contributions.
>
> If the authors are interested in my opinion beyond this review:
> Give this paper another iteration, remove PARAT completely, use the newly gained space to strengthen, especially the clarity of, the theoretical parts in the main paper + a convincing proof sketch for 4.5. If you feel like PARAT needs to be included, it needs to be clearer why, as this was also a concern in other reviews.
>
> Given such improvements, which I feel go beyond the reasonable adjustments of a rebuttal phase, I clearly see this paper a worthy contribution!

---

> ### Author Response · Authors · 2024-11-23
> **Thanks for your response & continued response to concerns [part 4]**
>
> Thank you for your timely response and acknowledging the contributions of our paper! It is true that we haven’t fully completed our response or updated the paper at the time of posting the previous replies. We wanted to post our partial written responses ASAP to facilitate timely discussion. We have updated the paper implementing the above mentioned changes now and attached our response to your final question in this response.
>
> Regarding the first part of your response "What do you mean by "non-uniform" here", in section 1.1 of our response, we explained the concept of non-uniform as having different circuits for different input lengths. This is also a well-defined term in circuit complexity [9]. Regarding the claim that the short statement of “Transformers can solve 3-SAT” oversells, in the previous response 1.2 we explained how this form of expression is used in the expressions of many precious seminar works in Transformer expressiveness, but we agree that this may cause misunderstandings for readers regarding the exact contributions. Correspondingly, we have adjusted to expression to “construct a non-uniform class of decoder-only Transformers that can solve 3-SAT using….”
>
> Again, we’re very grateful for pointing out important suggestions for our paper. Regarding your suggestions to remove PARAT completely, we fully agree that the engineering-focused PARAT compiler may not fully fit the theory-focused contributions of this paper and could instead be included in further work focusing on empirical contributions. However, the required structural changes may be too significant for the discussion period in the review phase. Instead, we will reduce the proportion focused on details of the compiler and add lemmas describing core theoretical contributions from the proof. Please see our updated manuscript for our efforts in this regard.
>
> Please let us know if our responses have addressed your concerns and if you have any further questions. If possible, we would be very glad if you can update the score reflecting any updated evaluation regarding the contributions of our paper.
>
> ## Q3 Definition of PARAT
>
> We directly answer your question with the following definition:
>
> **PARAT Language**
>
> The syntax of the PARAT language is a restricted subset of Python with the NumPy library. Every variable $v$ in PARAT is an 2-D Numpy array of shape $n\times d_v$, where $n$ denotes the input number of tokens and $d_v$ is the dimension of PARAT variable $v$ that can be different for every variable $v$.
>
> A pragram in the PARAT language is composed of a linear sequence of statements (i.e. no control flow allowed such as loops and branching), where each statement assign the value of an expression to a variable. Let `v_1`, `v_2`, … denote PARAT variable names, then each statement involving PARAT variables must be one of the following:
>
> - Binary operations: `v_1 + v_2` , `v_1 * v_2`, `v_1 - v_2`
> - Index operations: `v_1[v_2, :]` , `v_1[:, start:end]` for non-negative integers `start` and `end`
> - A function in our predefined library that takes as input PARAT variables
>
> The input variables of a PARAT program for vocabulary size $V$ are `tokens` and `indices` , where `tokens` is a $V$ dimension PARAT variable including one-hot token embeddings of the input tokens and `indices` is a 1 dimension PARAT variable s-op  including the numerical index of each input token (i.e. the array `[[1], [2], …, [n]]`)
>
> **ParametricTransformer Compiler**
>
> The ParametricTransformer **compiler** takes in a program written in the PARAT language and a PARAT variable `out` of dimension $V$ and outputs a PyTorch Module object that implements a Transformer model as defined in Section 2 such that the following is satisfied: For any possible input sequence of tokens $s$ in the vocabulary,  the token predicted by the Transformer model is the same as the token corresponding to `out[-1, :].argmax()`  (i.e., the token prediction at the last position) when interpreting the PARAT program with the Python interpreter with the NumPy library
>
> [9][https://en.wikipedia.org/wiki/Circuit_complexity#:~:text=Boolean circuits are one of,for all possible input lengths](https://en.wikipedia.org/wiki/Circuit_complexity#:~:text=Boolean%20circuits%20are%20one%20of,for%20all%20possible%20input%20lengths).

---

> > ### Comment · Reviewer_6FmX · 2024-11-27
> >
> > Thank you! I appreciate your effort in revising you original submission!
> >
> > I clearly see the improvement now regarding the role of PARAT, which I think gives the paper a much clearer focus. Now, the describtion PARAT feels like a nice complement to the overall theoretical (and expimental) results.
> >
> > Personally, I think the submission is now in a form that is near the level of acceptance. But, I think the newly introduced proof sketch and the overall wording in the paper still need a major revision, which feels beyond whats doable in a rebuttal phase.
> >
> >
> > Accordingly, I increase my score to '5' and want to reiterate my appreciation of the contribution the authors make.
> > Additionally, I see from other reviews that "transformer compiler" is an active field of research (see Tracr), which I was not aware of. Thus, I also decreased my confidence to '2' to highlight that I have no expertise aside from theoretical contributions made in the paper.

---

### Official Review · Reviewer_PKf5 · 2024-10-30

**Soundness:** 3
**Presentation:** 2
**Contribution:** 2
**Rating:** 5
**Confidence:** 3

**Summary:**

This paper provides a theoretical analysis of the Transformer's ability to solve 3-SAT problems through chain of thoughts, along with a compiler that can output a Transformer model satisfying a given specification. The empirical results demonstrate that the Transformer can achieve high accuracy in solving small-scale 3-SAT problems.

**Strengths:**

The problem tackled in this paper is highly interesting and relevant. I appreciate the effort of theoretical analysis conducted on the reasoning capabilities of transformer models, which is a rapidly evolving and impactful area of AI research.

**Weaknesses:**

This paper has several issues that need to be addressed.

First, the paper's introduction claims to answer a "fundamental question about whether LLMs can perform propositional reasoning." However, after reviewing the paper, it does not appear that the theoretical and empirical analysis provided is sufficient to conclusively answer this broad question. The analysis is focused narrowly on 3-SAT problems, and it is unclear how the results could be generalized to other propositional reasoning problems. Additionally, the model used in the experiments may not be representative of a typical large language model with billions of parameters.

Second, the proofs for the theorems presented in Section 4 are missing. While it is possible the proofs require additional space, it is necessary to at least provide a high-level outline of the proof approach in Section 4. Purely moving numerous lemmas and proofs to an appendix is an unsatisfactory way to present such a crucial part of the paper.

Third, the incremental contribution of the proposed compiler compared to the prior Tracr compiler is unclear. It appears that the primary focus of this work is on addressing implementation issues with Tracr, but the specific novel aspects are not well articulated. The paper should clearly explain what new capabilities or improvements are offered by the proposed compiler over the previous work.

**Questions:**

What is the incremental contribution of the proposed compiler compared to the prior Tracr compiler?

---

> ### Author Response · Authors · 2024-11-25
> **Thank you for your review & response to concerns**
>
> Thank you for your detailed review of the paper and insightful suggestions.
>
> ## Include lemmas in the main paper
>
> Thank you so much for pointing out the importance of including important lemmas in the main paper to emphasize our theoretical contribution. We fully agree with this statement and, after careful consideration of your and other reviewers’ feedback, decide that the details of the PARAT compiler are less important for this theory-focused work and instead should include more details on our theoretical contribution. In this regard, we included multiple definitions and lemmas that were extracted from our main proof and included in the main text. Please see our updated PDF for related changes. We’ll be very grateful if you can provide further comments and feedback on this updated section.
>
> ## Fundamental question about whether LLMs can perform propositional reasoning
>
> Thank you for pointing out that our statement of “propositional reasoning” is not exactly the same term as 3-SAT. However, we are slightly confused by the statement since SAT-Solving is equivalent to formal propositional reasoning. In the paper“Propositional Reasoning by Model”, which introduced the concept, propositional reasoning is defined as “deduction depending on *if, or, and,* and *not*”, which is encompassed by SAT. SAT-Solving is also linearly equivalent to 3-SAT, as demonstrated by the Tseitin Transformation, which shows that and SAT formula can be efficiently converted to an equivalent 3-SAT formula that’s linear in the original size. Therefore, our claim that our paper answers “a fundamental question about whether Transformers can perform propositional reasoning” is well-justified under the above equivalences. However, the sentences may indeed be unclear in meaning without further explanation of the concepts, and we have changed the statement to “a fundamental question about whether Transformers can perform propositional reasoning with the 3-SAT problem.”
>
> ## Advantages compared to the Tracr Compiler
>
> Thanks for pointing out that we haven’t explicitly stated the advantages of our compiler compared with the Tracr Compiler. The advantage of our compiler lies in increase expressiveness and reduced parameter complexity, which makes our compiler much more suitable for implementing complex algorithms with large input size.
>
> In particular:
>
> - **Expressiveness:** The PARAT language supports operations on 2-D arrays of size $n\times d_v$ where $d_v$ is a dimension specific to variable $v$ and $n$ is the input length. The RASP language for which Tracr is based on only supports operations on 1-D arrays of size $n$. Therefore, PARAT supports a more diverse set of operations than RASP. For example, the operation for checking satisfaction and detecting conflict can both be implemented with a single PARAT statement but have no simple implementation in RASP that we know of.
> - **Parameter Complexity w.r.t. Context Length:** The ParametricTransformer Compiler generates Transformers uses a single embedding dimension to embed the position of each token, and the dependence of parameter number (excluding positional encodings) on the context length is $O(1)$. In contrast, the Tracr compiler uses $O(l)$ embedding size to create a one-hot encoding of each position for context length $l$ and results on $O(l^2)$ number of Parameters.
> - **Parameter Complexity for Arithmetic Operations:** For basic binary arithmetic operations such as addition, subtraction, and multiplication of 1-D variables, a ParametricTransformer compiled model requires only $O(1)$ additional parameters. However, the Tracr compiler enumerates every possible value combination of the 2 operands, and as such requires $O(v^2)$ parameters for parameters that can take up $v$ possible values.
>
> Please let us know if you concerns have been addressed and if you have further questions or comments.

---

> > ### Comment · Reviewer_PKf5 · 2024-11-26
> >
> > Thank you for the authors' response! After careful consideration, I still have concerns regarding the presentation of the theoretical analysis and the overall contribution of the paper. As such, I need to maintain my current score at this time. Regardless of whether the paper is ultimately accepted or not, I strongly recommend that the authors refine the paper presentation to enhance clarity and improve the overall quality of the work.

---

### Official Review · Reviewer_PEeH · 2024-11-04

**Soundness:** 2
**Presentation:** 3
**Contribution:** 1
**Rating:** 5
**Confidence:** 3

**Summary:**

This paper investigates the logical reasoning capabilities of Transformer models in solving SAT problems. The authors designed a decoder Transformer to emulate the DPLL algorithm, tackling SAT through variable selection, conflict detection, and backtracking. The PARAT compiler translates high-level procedural logic into model weights, enabling practical implementation.

**Strengths:**

1. Leveraging a Transformer to simulate the entire DPLL SAT-solving process, rather than focusing on optimizing specific heuristics such as variable selection, represents a novel approach.
2. The paper presents the PARAT compiler, which translates the procedural logic of the DPLL solving process into a Transformer model, achieving perfect accuracy on SAT problems with up to 20 variables and 88 clauses.
3. The trained model sustains high accuracy on SAT test instances with variable counts comparable to those in the training dataset.

**Weaknesses:**

1. The evaluation assessed only the accuracy of satisfiability predictions, without examining the correctness of the reasoning process itself. Thus, even when predictions were correct, the reasoning steps might still diverge from the intended algorithm. Additionally, there was no in-depth analysis of the accuracy drop on larger SAT instances to identify if errors arose from issues in conflict analysis, backtracking, or other specific areas.
2. The experiments compared the Chain-of-Thought (CoT) length solely with theoretical upper bounds, without including baseline comparisons. Evaluating against the reasoning steps of a standard DPLL solver or other baselines could offer a clearer perspective on the model’s reasoning capabilities.
3. The trained model shows limited generalization, performing well only on test instances with variable counts similar to those in the training data. Its accuracy declines significantly on SAT instances with up to 20 variables, possibly dropping below 50%.

**Questions:**

The model demonstrates poor generalization, achieving high accuracy only on SAT instances with variable counts similar to those in the training data. Performance significantly declines on larger instances, indicating that it may be learning patterns specific to the training data rather than pure logical reasoning, thus failing to fully substantiate the claim that Transformers are capable of logical reasoning.

---

> ### Author Response · Authors · 2024-11-22
> **Thank you for your review & Response to concerns**
>
> Thank you for your detailed review of the paper and pointing our possible improvements for our current version of the paper. We will discuss your concerns in detail in the following section:
>
> ## Limited Length Generalization is supportive of our theoretical results and experimental conclusions
>
> We fully agree with the statement that the trained Transformer models demonstrate limited length generalization, and this phenomenon is **supportive of our arguments in the paper**. First, we would like to clarify that our core contribution is the theoretical result as stated in Theorem 4.5, and our experiments provide supplemental evidence on the capabilities of Transformers trained on the chain of thought used by our theoretical construction. Our theoretical result shows that, for any bounded length, a Transformer can decide all SAT instances within that length. However, the model weights need to be adjusted to account for longer-length inputs. Therefore, the experimental result that trained Transformers perform nearly perfectly within the training lengths but fail to length-generalize exactly fits our theoretical predictions, which is mentioned in lines 537 and 538. The limitation of trained Transformers in length generalization is explicitly stated as one of our experimental conclusions, as demonstrated by section 6.2 titled “limitation in length generalization”. As such, we believe that “Its accuracy declines significantly on SAT instances with up to 20 variables, possibly dropping below 50%” should not be considered a weakness of our work and is instead in support of both our theorems and our experimental conclusions.
>
> ## Reasoning Process Evaluation
>
> Thank you for pointing out that the reasoning process of the model may be erroneous even though the final SAT vs UNSAT prediction is correct. We fully agree that this is theoretically possible for purely random 3-SAT instances, and models may guess the correct SAT vs UNSAT solution using statistical properties without performing correct reasoning. However, we design our marginal evaluation dataset such as the SAT and UNSAT instances in the evaluation set differently by a single token. Therefore, the statistical properties between SAT vs UNSAT instances are in-differentiable (lines 424-427), and it is impossible for trained Transformers to achieve near-perfect accuracy without the capability for logical reasoning. Therefore, we do not believe that “even when predictions were correct, the reasoning steps might still diverge from the intended algorithm” weakens our experimental arguments on the intra-length out-of-distribution generalization capability of Large Language Models.
>
> ## Baseline Comparisons and In-depth Error Analysis
>
> Thank you for suggesting further experimental comparisons and analysis to better understand our model capabilities. We agree that these additional experiments may provide further insights into our trained and compiled models. However, we do not see a clear connection between the suggested experiments and our main arguments.
>
> We would like to reiterate that our core contribution is theoretical (theorem 4.5), which demonstrates that Transformers are capable of solving the NP-hard logical reasoning problem 3-SAT, and the experiments provide supplemental evidence to validate our theoretical results. Towards this end, we performed experiments to demonstrate that the theoretical upper bound indeed holds in the implementation of our theoretical construction, and demonstrates that trained Transformer models achieve near-perfect intra-length OOD generalization but have limitations in length generalization.
>
> Therefore, we believe that providing an in-depth error analysis of the trained Transformers does not directly provide additional evidence for our theoretical results and may distract readers following the main argument flow since our theoretical argument does not concern failure modes of Transformer models. We also do not intend to compete with practical SAT-solvers and instead focus on whether Transformers can perform correct propositional reasoning, as stated in the contributions section. Therefore, comparing with a standard DPLL does not seem to contribute to our core argument.
>
> Again, we are very thankful for your insightful and detailed reviews and would like to clarify potential misunderstandings of our paper. Please let us know whether our explanations suffice to address your concerns and if you have further comments and we gladly welcome further discussions.

---

### Official Review · Reviewer_fauL · 2024-11-04

**Soundness:** 2
**Presentation:** 2
**Contribution:** 1
**Rating:** 3
**Confidence:** 3

**Summary:**

This work investigates the application of the Transformer architecture for SAT solving.
To this end, the authors present theoretical boundaries on decoder-only Transformers' size and solving capabilities in 3-SAT problems.
Furthermore, they introduce "Parametric Transformer" (PARAT) to create and evaluate the SAT-solving model.
The evaluation itself consists of a simulation study where 3-SAT problems with varying numbers of variables have been generated as a benchmark.
The authors show how their compiled model is able to solve all problems with perfect accuracy, while models that were trained instead fail when challenged with problems of previously unseen numbers of variables.

**Strengths:**

The authors present how a 3-SAT solving algorithm can be expressed within the confinements of the building blocks of the Transformer architecture and demonstrate that it remains accurate across varying numbers of involved variables and clauses.

**Weaknesses:**

The contribution and significance of PARAT remained unclear to me after considering both the main text and appendices.
The authors claim that PARAT "is designed to provide an intuitive syntax resembling standard numerical array manipulation, akin to NumPy, [...]" (page 6, line 311).
Considering the referenced "ParametricTransformer Code" in Appendix D, it becomes evident that the language in question is pure Python using the numpy library.
The language "Python" is named not once in the paper, and the formulation of contributing a "compiler" with the notation "akin to" numpy and PARAT's "NumPy-like Array Syntax" (page 6, line 314) seems completely unjustified and bordering on deception.

**Questions:**

I have no questions for the authors.

---

> ### Author Response · Authors · 2024-11-23
> **Thank you for your review and response to concerns**
>
> We greatly appreciate the effort you put into reviewing the paper and thank you for pointing out a potential point of confusion regarding the exact contributions of PARAT. We acknowledge that our current description of PARAT contains informalities that may lead to confusion regarding our contributions. Therefore, we would like to clarify the potential misunderstanding indicated in your review regarding the purpose of the code samples we provided and the PARAT compiler.
>
> Our main contribution in the PARAT section is the compilation process of array operations expressed in PARAT into Transformer model weights. Therefore, the resemblance of PARAT code with NumPy operations does not render our contributions trivial. PARAT is designed to resemble a “NumPy-like array manipulation syntax”; therefore, many code statements we provide in the paper and appendix are also valid NumPy operations on arrays. However, the purpose of the compiler ParametricTransformer is not to execute the code on given inputs. Instead, the provided PARAT code serves as the input to the compiler, and the compiler produces a Transformer model in PyTorch as output. This output model implements the same computations as those described in the input code when the model takes in a sequence of tokens as input.
>
> Therefore, we respectfully disagree with the claim that our contribution of PARAT is “akin to deception” and would like to clarify our intent. While the implementation of the PARAT language and compiler is based on Python, we believe this was implicitly clear given the context and the references to PyTorch and NumPy. Python has been the de facto programming language for most code associated with deep learning. Furthermore, we explicitly stated that our compiler “outputs a PyTorch model” and that the syntax of PARAT is designed “akin to NumPy,” both of which are well-known Python libraries.
>
> Additionally, the main contribution of the paper is the theoretical result that Transformers can solve SAT. Both the compiler and the experiments provide supplemental evidence to verify several predictions of our theorems. We hope that the reviewer finds the major conclusions of our work compelling, as they represent the primary contributions of our paper.
>
> Thank you again for your review. We are happy to address any further questions you may have.

---

### Official Review · Reviewer_DcJi · 2024-11-05

**Soundness:** 4
**Presentation:** 4
**Contribution:** 4
**Rating:** 8
**Confidence:** 3

**Summary:**

This paper investigates the capabilities of LLMs to solve an NP-complete problem, the 3-SAT problem. The authors first prove by construction that LLMs can simulate the DPLL algorithm to solve the 3-SAT problem with a bounded number of variables. To support this theoretical argument, they also compiled a Transformer implementation from the construction, which perfectly solved the bounded 3-SAT problems in their evaluation. A further empirical investigation demonstrated that, instead of by construction, LLMs could learn to solve the bounded 3-SAT problems whereas they are unable to generalize to problems with more variables. This observation matches the limitation of the theoretical construction requiring a bounded number of variables for 3-SAT problems.

**Strengths:**

1. This is the first work to study the capabilities of LLMs to solve NP-complete problems.
2. This paper proves by construction that LLMs can simulate the DPLL algorithm to solve the 3-SAT problem with, though, a bounded number of variables. It demonstrates that Transformers can perform unit propagation and backtracking, which are the core operations of the DPLL algorithm.
3. To support the theoretical argument, the authors also developed a compiler to compile the theoretical construction into a Transformer implementation and verified its practical capability to perfectly solve bounded 3-SAT problems.
4. The evaluation results match the theoretical statement. The main theorem implies that Transformers can perfectly solve 3-SAT problems with a bounded number of variables. In the evaluation, they also observed near-100% accuracy over problems with a bounded number of variables whereas the accuracy dropped significantly as the variable number further increased.

**Weaknesses:**

1. The main theorem implies that an arbitrarily large LLM is required to simulate a general DPLL algorithm for solving all 3-SAT problems. Considering a bounded program (a program of a specific size) can implement the general DPLL algorithm, it is still unclear how to approach the general DPLL algorithm using a bounded LLM.
2. The experiment setting in Sec 6.2 contradicts the description of Figure 2. Sec 6.2 states that the model is trained on the Marginal dataset and evaluated on the three generated datasets while Figure 2 describes the other way.

**Questions:**

1. What is the maximum number of steps, w.r.t. the variable number, for running DPLL over the generated 3-SAT problems? How does it compare to the empirical maximum CoT length?
2. What is the accuracy of the compiled model on instances with more than 20 variables, i.e. beyond the guaranteed upper bound? Does the accuracy drop drastically?

---

> ### Author Response · Authors · 2024-11-27
>
> We’re greatly thankful for your acknowledgement of the strengths and contributions of the paper. Also, thanks for pointing our the description inconsistencies between the figure and section 6.2. The Figure description is accurate (i.e. Trained on different distributions and evaluated on marginal only), and we have fixed section 6.2 accordingly.
>
> For the 2 questions you posted, both of them are insightful and important questions regarding our paper. However, we find them difficult to answer directly due to several technical details. Therefore, we will first explain the technical context related to these questions and provide the closest answer to your question according to our understanding.
>
> Finally, we would greatly appreciate if you could provide some comments on the discussion threads of other reviewers.
>
> **1. DPLL CoT Length**
>
> Technical context:
>
> - The **number of steps** of the DPLL algorithm is not a well-defined concept, even though this can be easily defined as the number of Chain-of-Thought tokens for a decoder Transformer model. Our theoretical construction uses Chain-of-Thought to simulate various high-level operators performed by the DPLL algorithm, as summarized by abstract DPLL state transition system, but even so each transition may be simulated by more than one token in the Chain-of-Thought
>     - We believe what you’re referring to in the question is the number of tokens of a Chain-of-Thought that trace-simulates a concrete run of the DPLL algorithm they way illustrated in Figure 1 in the paper
> - **Dependency on branching heuristic** The performance of the DPLL algorithm is heavily dependent on a branching heuristic function, which selects decision literal (i.e., the variable picked to assume a truth value) from all unassigned variables when no deduction is possible. Therefore, there is no exact number of steps for the general DPLL algorithm without a specific heuristic. Our theorem proves that our theoretical construction’s CoT is trace equivalent to an DPLL algorithm with a particular heuristic (approximation of the Maximum Occurrences on clauses of Minimum size (MOM’s) heuristic).
>     - Therefore, to answer the first part of your question
>
> With the above assumption, here we present the maximum number of tokens for a DPLL algorithm with the Jeroslow-Wang (JW) Heuristic vs our compiled model:
>
> Marginal:
>
> | Number of Variables | 4 | 5 | 6 | 7 | 8 | 9 | 10 | 11 | 12 | 13 | 14 | 15 | 16 | 17 | 18 | 19 | 20 |
> | --- | --- | --- | --- | --- | --- | --- | --- | --- | --- | --- | --- | --- | --- | --- | --- | --- | --- |
> | DPLL with JW | 31 | 40 | 54 | 59 | 66 | 89 | 102 | 122 | 125 | 151 | 221 | 170 | 194 | 229 | 257 | 269 | 341 |
> | Theoretical Construction | 37 | 45 | 49 | 65 | 91 | 84 | 118 | 125 | 146 | 185 | 160 | 199 | 211 | 263 | 338 | 330 | 411 |
>
> Random:
>
> | Number of Variables | 4 | 5 | 6 | 7 | 8 | 9 | 10 | 11 | 12 | 13 | 14 | 15 | 16 | 17 | 18 | 19 | 20 |
> | --- | --- | --- | --- | --- | --- | --- | --- | --- | --- | --- | --- | --- | --- | --- | --- | --- | --- |
> | DPLL with JW | 31 | 48 | 56 | 62 | 74 | 82 | 90 | 98 | 123 | 165 | 182 | 168 | 181 | 219 | 215 | 302 | 352 |
> | Theoretical Construction | 31 | 42 | 54 | 59 | 72 | 106 | 93 | 148 | 160 | 152 | 206 | 197 | 225 | 276 | 265 | 353 | 292 |
>
> Skewed
>
> | Number of Variables | 4 | 5 | 6 | 7 | 8 | 9 | 10 | 11 | 12 | 13 | 14 | 15 | 16 | 17 | 18 | 19 | 20 |
> | --- | --- | --- | --- | --- | --- | --- | --- | --- | --- | --- | --- | --- | --- | --- | --- | --- | --- |
> | DPLL with JW | 24 | 28 | 37 | 42 | 52 | 49 | 70 | 72 | 81 | 79 | 123 | 103 | 124 | 131 | 142 | 159 | 170 |
> | Theoretical Construction | 31 | 30 | 36 | 44 | 58 | 50 | 72 | 65 | 99 | 86 | 101 | 109 | 148 | 131 | 160 | 155 | 163 |
>
> While not always the case, the DPLL with JW heuristic generally uses less corresponding CoT steps than our theoretical construction. This shows that the branching heuristic in our Theoretical Construction is still less efficient than the JW heuristic.
>
> **2. Accuracy on more than 20 variables**
> - Our compiled model for 20 variables only includes token embeddings for the 20 variable tokens. Therefore, if we were to attempt to solve a 21-variable formula, then it will not have the corresponding token embedding for variable 21 and will result in an error.
>
> Instead, we can investigate models with a sufficient number of variable tokens but designed for a smaller number of clauses(i.e., smaller $c$ value in Theorem 4.5). Different $c$ values correspond to different scaling factors for the attention weights, which is represented by $\beta$ and explained in the “Soft vs Hard Attention” paragraph in section 5.2. As such, we can investigate your proposed question by adjusting smaller $\beta$ values of a model designed for a larger number of variables and clauses. We have indeed investigated our compiled model from this aspect **in Figure 4**. In this figure, you may observe how models without sufficient $\beta$ parameter perform on formulas with different lengths.

---

> > ### Comment · Reviewer_DcJi · 2024-12-02
> >
> > Thanks for the clarification and additional results. I can see the empirical CoT lengths are close for DPLL and your compiled model, which are much smaller than your theoretical bound $p\cdot 2^{p+1}$.

---

### Meta-Review · Area_Chair_gjWa · 2024-12-20

**Metareview:**

This paper investigates how well LLMs can solve the 3-SAT problem. In particular, it constructs a decoder-only Transformer to solve SAT using backtracking and deduction via Chain-of-Thought (CoT). The reviews agree that this is overall an important question. Unfortunately, there is also much confusion here that should be clarified before publication. First, it is not true that this is the "first work to study the capabilities of LLMs to solve NP-complete problems," as one reviewer argued, as the following selected pointers to SAT works show:

https://github.com/casmlab/NPHardEval

Zhengyuan Shi, Min Li, Yi Liu, Sadaf Khan, Junhua Huang, Hui-Ling Zhen, Mingxuan Yuan, Qiang Xu: SATformer: Transformer-Based UNSAT Core Learning. ICCAD 2023: 1-4
https://arxiv.org/abs/2209.00953

Minyu Chen, Guoqiang Li, Ling-I Wu, Ruibang Liu, Yuxin Su, Xi Chang, Jianxin Xue:
Can Language Models Pretend Solvers? Logic Code Simulation with LLMs. SETTA 2024: 102-121
https://arxiv.org/html/2403.16097v1

Jundong Xu, Hao Fei, Liangming Pan, Qian Liu, Mong-Li Lee, Wynne Hsu:
Faithful Logical Reasoning via Symbolic Chain-of-Thought. ACL (1) 2024: 13326-13365

Second, in the feedback to one of the reviewers, the author argue that there is a potential misunderstanding that the paper  <<“investigates the application of the Transformer architecture for SAT solving,” whereas our primary focus is to “answer a fundamental question about whether LLMs can perform propositional reasoning,” which is a different>> To be honest, I do not see how this is different. 3SAT is NP-complete, and SAT solvers are equal to "propositional reasoning". That is also, I guess, why the abstract refers to 3SAT and SAT solvers. Finally, one reviewer points out that the "evaluation assessed only the accuracy of satisfiability predictions, without examining the correctness of the reasoning process itself." While I see the general argument of the authors provided in the rebuttal, adding such an experiment would improve the papers. Overall, this is a borderline case, and due to many open issues and missing related work, I recommend pushing for one of the next venues. Please note that this decision should not be taken as a statement regarding the usefulness of your research.

**Additional Comments On Reviewer Discussion:**

The discussion arose from issues raised in the reviews. There was a longer discussion around some of the issues raised where the reviewer also acknowledged to be a bit too strong in wording and soften the statement. I appreciate this. Due to the missing related work, however, the discussion was not that important for the overall decision.

---

### Decision · Program_Chairs · 2025-01-22

Reject